# No relationship between frontal alpha asymmetry and depressive disorders in a multiverse analysis of five studies

**Aleksandra Kołodziej[1]\*, Mikołaj Magnuski[1], Anastasia Ruban[1], Aneta Brzezicka[1,2]**

[1]University of Social Sciences and Humanities, Warsaw, Poland; [2]Cedars-Sinai Medical Center Department of Neurosurgery, Los Angeles, United States

**Abstract** For decades, the frontal alpha asymmetry (FAA) – a disproportion in EEG alpha oscillations power between right and left frontal channels – has been one of the most popular measures of depressive disorders (DD) in electrophysiology studies. Patients with DD often manifest a left-sided FAA: relatively higher alpha power in the left versus right frontal lobe. Recently, however, multiple studies failed to confirm this effect, questioning its reproducibility. Our purpose is to thoroughly test the validity of FAA in depression by conducting a multiverse analysis – running many related analyses and testing the sensitivity of the effect to changes in the analytical approach – on data from five independent studies. Only 13 of the 270 analyses revealed significant results. We conclude the paper by discussing theoretical assumptions underlying the FAA and suggest a list of guidelines for improving and expanding the EEG data analysis in future FAA studies.

## Introduction

Electrophysiological studies on frontal alpha asymmetry (FAA) in depressive disorders (DD) have almost 40 years of history, with first reports presented in 1983 (*Schaffer et al., 1983*). Since then, many studies have reported relatively higher alpha band power in the left vs right frontal channels (left-sided FAA) in subjects suffering from DD compared to healthy individuals (*Allen et al., 2004*; *Davidson, 1984*; *Davidson, 2004*; *Kemp et al., 2010*; *Schaffer et al., 1983*). FAA index, calculated by subtracting the left-side alpha power from the respective right-side channel, is one of the most common electrophysiological indicators of DD in the current literature (*de Aguiar Neto and Rosa, 2019*). However, multiple studies failed to replicate the relationship between FAA and DD (*Allen et al., 2004*; *Carvalho et al., 2011*; *Deldin and Chiu, 2005*; *Gold et al., 2013*; *Kaiser et al., 2018a*; *Kentgen et al., 2000*; *Knott et al., 2001*; *Mathersul et al., 2008*; *Szumska et al., 2021*; *Vuga et al., 2006*) and conclusions of meta-analyses remain skeptical (*Thibodeau et al., 2006*; *van der Vinne et al., 2017*). In this light statements about FAA being a biomarker of depression (*Baskaran et al., 2012*; *Iosifescu et al., 2009*) seem to be too far-fetched.

It is not clear what the causes of above-mentioned inconsistency in the literature are, but methodological issues are mentioned as one potential problem in a recent review by *Kaiser et al., 2018b*. Factors like age, gender, or education are known to covary with depression (*McFarland and Wagner, 2015*; *Nolen-Hoeksema, 2001*; *Stordal et al., 2003*) and alpha power or asymmetry (*Jesulola et al., 2017*; *Parameshwaran and Thiagarajan, 2019*; *van der Vinne et al., 2017*), but are often not controlled for during recruitment. While counter-balancing groups with respect to these variables can be difficult their effect may be accounted for by including them as predictors in the regression model. Although correlated predictors like education and depression can reduce each other's effect by explaining shared variance in the dependent variable, including confounding

\*For correspondence:
akolodziej@swps.edu.pl

**Competing interests:** The authors declare that no competing interests exist.

variables can also help remove variance unexplained by the variable of interest and therefore increase its effect.

There are also other important methodological problems worth considering when trying to resolve the validity of FAA in DD. Although much attention in the FAA literature has been paid to the choice of EEG reference (see for example: *Smith et al., 2017*; or *Stewart et al., 2014*) other aspects of signal processing and analysis seem to be more neglected. Many EEG studies on FAA use and report FAA index calculated only for a few channel pairs (e.g. one or two pairs were used in 12 out of 17 studies [70.6%] included in the meta-analysis by *van der Vinne et al., 2017*). In combination with the fact that topographical maps of effects are rarely presented (4/17 studies [23.5%] in *van der Vinne et al., 2017*) this significantly reduces the reliability and interpretability of the reported effects. The FAA effects are frequently assumed to reflect frontal sources of alpha oscillations but without a topographical map to support this claim it is difficult to conclude whether such interpretation is correct. For example, alpha asymmetry at frontal channels may, in principle, arise due to asymmetrical projection from other, non-frontal, sources. This could be identified in the topography, but not at the single channel pair's level. Without a topographical map it is also more difficult to assess the physiological reliability of the reported effect – significant effect on one channel pair without similar effects on surrounding channels calls for skeptical consideration (*van Ede and Maris, 2016*). Therefore, it might be better to perform the analysis on many frontal channel-pairs with relevant correction for multiple comparisons (for example, with the very popular cluster-based permutation approach, *Maris and Oostenveld, 2007*). This is unfortunately rarely done in FAA studies on DDs (0/17 studies in *van der Vinne et al., 2017*).

However, performing the analysis only at the channel level, when the research question pertains to the neural source of the effects, can also lead to misinterpretations. Even if topographies are shown, they can be inconclusive with respect to the underlying neural source. For this reason it might be useful to perform source localization and continue the analyses in the source space. Given the assumption of frontal alpha sources of FAA presented in the literature, source level analysis would be appropriate. Regrettably, most FAA studies do not perform source localization, although there are notable exceptions (for example, *Lubar et al., 2003*; *Smith et al., 2018*).

Incompatible results in FAA literature, summarized briefly above, suggest that the FAA relationship with DD is sensitive to the choice of signal preprocessing and analysis steps. In such a case applying multiverse analysis (*Steegen et al., 2016*), that is, presenting results of multiple justifiable analysis paths, is a valuable tool to test the robustness of the studied effects. Multiverse analysis seems to be especially well suited for neuroscience research, given the multitude of preprocessing and data analysis choices that result in a complex 'garden of forking paths' (*Gelman and Loken, 2014*). As most neuroscience studies test only one analysis variant it is difficult to assess the robustness of any individual effect, and it seems that at least some neuroscientific findings are sensitive to the choice of signal analysis steps (*Cohen, 2015*; *Cohen and Gulbinaite, 2014*; see also *Botvinik-Nezer et al., 2019*).

The purpose of this article is to thoroughly test the robustness and credibility of FAA as a marker of DDs and address the limitations of FAA research methodology by performing a multiverse analysis of data coming from five independent studies. We performed 270 analyses in total differing in: (a) the signal space used (channel space vs source space); (b) subselection of the signal space (channel pairs vs all frontal pairs with cluster-based correction); (c) statistical contrast used (group contrasts vs linear regression); (d) statistical control for confounding variables (gender, age and education). Finally we perform analyses on data aggregated across studies and propose additional guidelines to improve quality and reliability of the data analysis in FAA research.

## Results

To investigate the validity and robustness of FAA as a marker of DDs we used the multiverse approach (*Steegen et al., 2016*) and performed a total of 270 analyses of eyes-closed resting EEG recordings (total N = 388) from five independent studies. These data sets differ in EEG recording equipment, cap layout, and characteristics of subject groups (see *Figure 1* and *Materials and methods*, sections: *Participants* and *Electrophysiological data sets*).

The analysis variants making up the multiverse analysis can be classified along four major dimensions (*Figure 2*): (a) statistical contrast used: group comparisons or testing for a linear relationship,

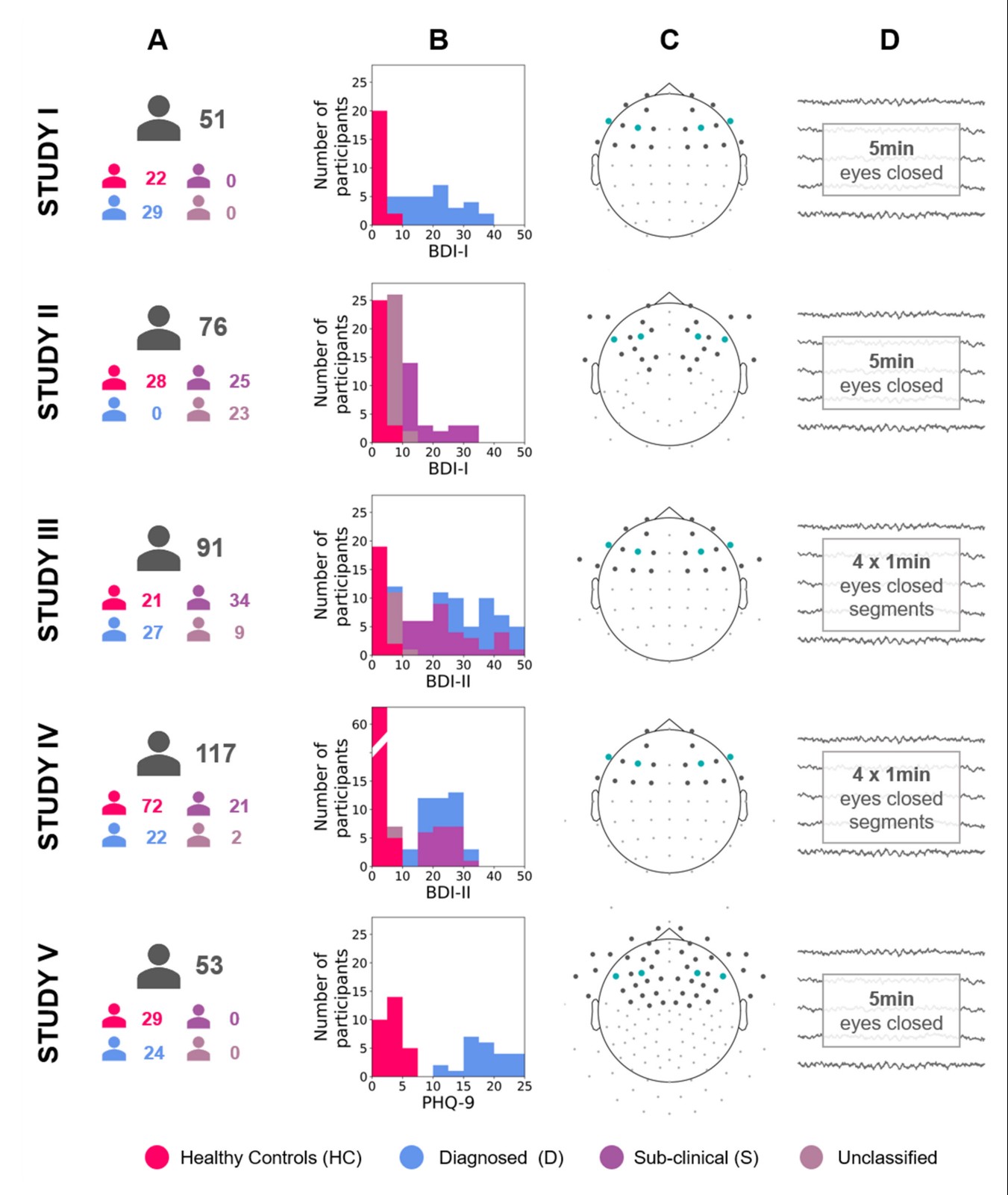

**Figure 1.** Diagram describing the five studies included in this article (Studies I, II, III, IV, and V). (**A**) Number of participants for each study and group (see *Table 12* for details). (**B**) Stacked histograms showing the distribution of BDI or PHQ-9 scores in each study and each group. (**C**) Channel montage. Frontal channels used in cluster-based analyses are marked with gray dots. Channels used in channel-pairs analysis are marked with teal dots (F3–F4, F7–F8, and corresponding channels in the EGI montage). (**D**) Rest period length and scheme.

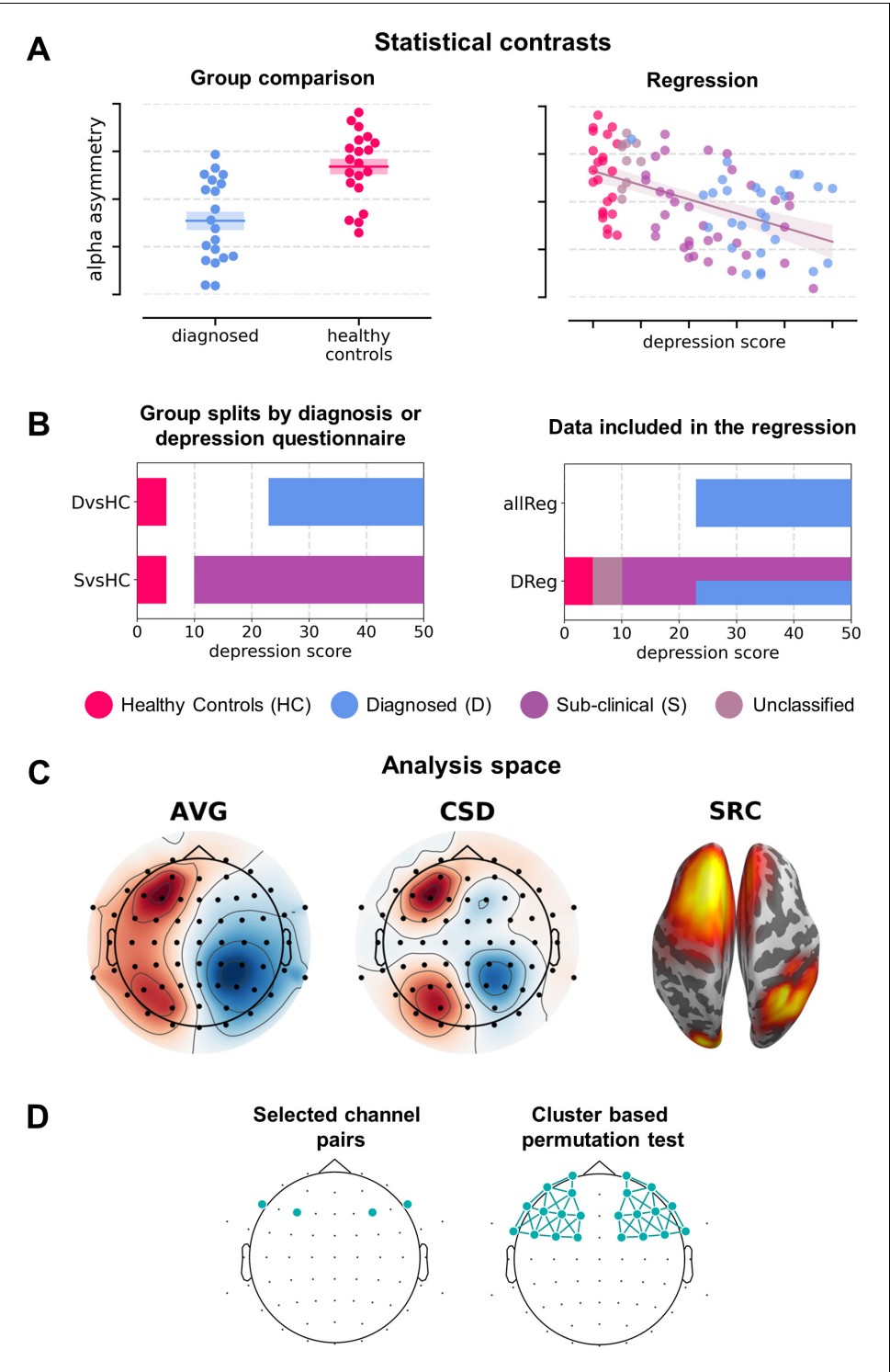

**Figure 2.** Analysis variants used (described in detail in *Variants of statistical analysis* section). (**A**) Schematic depiction of given statistical contrast: group comparisons (left) vs regression (right). (**B**) Specification of each contrast against depression scores. Left panel shows a schematic range of depression scores for each contrast: diagnosed vs healthy controls (*DvsHC*) and sub-clinical vs healthy controls (*SvsHC*). Right panel shows the range of depression scores for data included in each linear contrast: regression on diagnosed subjects (*DReg*) uses only subjects with clinical diagnosis, while regression on all subjects (*allReg*) uses all subject groups. The color legend for the subject groups is presented below these figures. (**C**) Analysis space: AVG – channel level, average reference; CSD – channel level, current source density; SRC – source level, DICS beamforming. (**D**) Schematic depiction of analysis method: selected channel pairs versus all frontal channels with cluster-based correction for multiple comparisons.

(b) the signal space used: channel space (average reference – AVG: 120 analyses, 44%; current source density reference – CSD: 120, 44%) or source space (DICS beamforming, 30, 11%); (c) subselection of the signal space: channel pairs (120, 44%), all frontal pairs with cluster correction (60, 22%) or all frontal channels with cluster-based correction and standardization instead of subtraction (60, 22%; see *Signal analysis* section); and (d) statistical control for confounding variables (135 without and 135 with control for confounds).

We used four different statistical contrasts in the analyses: two group contrasts using independent t-tests to compare FAA between groups; and two linear contrasts using linear regression to

**Table 1.** Results for all channel-pair analyses.

Each row represents two channel-pair results for a given contrast, study, and space combination; uncorrected for multiple comparisons. Electrode placement for each study is shown in *Figure 1C*. (*N*: number of participants included in given contrast; *ES*: effect size; Cohen's d for group comparison and Pearson's r for regression; *CI*: bootstrap 95% confidence interval for the effect size).

| | | | | | Selected electrodes without correction | | | | | | | |
| | | | | | Pair 1 (F3–F4) | | | | Pair 2 (F7–F8) | | | |
| No. | Contrast | Study | Space | N | t | p | ES | CI | t | p | ES | CI |
|---|---|---|---|---|---|---|---|---|---|---|---|---|
| 1 | DvsHC | I | avg | 29 vs 22 | −2.073 | 0.043 | −0.573 | [−1.135,−0.024] | −0.365 | 0.717 | −0.101 | [−0.644, 0.465] |
| 2 | DvsHC | I | csd | 29 vs 22 | 0.132 | 0.896 | 0.038 | [−0.550, 0.608] | 0.553 | 0.583 | 0.153 | [−0.415, 0.689] |
| 3 | DvsHC | III | avg | 27 vs 21 | 0.904 | 0.371 | 0.247 | [−0.316, 0.689] | −0.536 | 0.595 | −0.145 | [−0.760, 0.452] |
| 4 | DvsHC | III | csd | 27 vs 21 | 0.849 | 0.401 | 0.226 | [−0.307, 0.721] | −0.129 | 0.898 | −0.035 | [−0.590, 0.536] |
| 5 | DvsHC | IV | avg | 22 vs 72 | 0.450 | 0.654 | 0.094 | [−0.310, 0.510] | 0.277 | 0.783 | 0.059 | [−0.413, 0.463] |
| 6 | DvsHC | IV | csd | 22 vs 72 | 0.767 | 0.449 | 0.212 | [−0.287, 0.771] | −1.396 | 0.172 | −0.345 | [−0.808, 0.167] |
| 7 | DvsHC | V | avg | 24 vs 29 | 2.823 | 0.007 | 0.743 | [0.218, 1.255] | 1.927 | 0.061 | 0.501 | [−0.023, 0.998] |
| 8 | DvsHC | V | csd | 24 vs 29 | 0.748 | 0.458 | 0.208 | [−0.380, 0.775] | −0.727 | 0.471 | −0.202 | [−0.766, 0.370] |
| 9 | SvsHC | II | avg | 23 vs 28 | 0.201 | 0.841 | 0.056 | [−0.502, 0.614] | −0.662 | 0.511 | −0.179 | [−0.780, 0.381] |
| 10 | SvsHC | II | csd | 23 vs 28 | 1.144 | 0.258 | 0.318 | [−0.221, 0.827] | −0.199 | 0.843 | −0.054 | [−0.611, 0.508] |
| 11 | SvsHC | III | avg | 33 vs 21 | −0.852 | 0.398 | −0.209 | [−0.640, 0.306] | −1.328 | 0.190 | −0.332 | [−0.804, 0.216] |
| 12 | SvsHC | III | csd | 33 vs 21 | −1.181 | 0.244 | −0.280 | [−0.730, 0.219] | −1.094 | 0.280 | −0.302 | [−0.798, 0.219] |
| 13 | SvsHC | IV | avg | 21 vs 72 | 1.359 | 0.184 | 0.346 | [−0.198, 0.816] | 1.247 | 0.219 | 0.254 | [−0.138, 0.646] |
| 14 | SvsHC | IV | csd | 21 vs 72 | 0.558 | 0.581 | 0.147 | [−0.393, 0.655] | 0.141 | 0.889 | 0.035 | [−0.388, 0.605] |
| 15 | allReg | I | avg | 54 | −1.138 | 0.260 | −0.156 | [−0.397, 0.088] | −0.180 | 0.858 | −0.025 | [−0.394, 0.258] |
| 16 | allReg | I | csd | 54 | −0.540 | 0.591 | −0.075 | [−0.309, 0.167] | 0.077 | 0.939 | 0.011 | [−0.335, 0.280] |
| 17 | allReg | III | avg | 91 | 0.545 | 0.587 | 0.058 | [−0.105, 0.263] | −0.781 | 0.437 | −0.083 | [−0.271, 0.101] |
| 18 | allReg | III | csd | 91 | 0.209 | 0.835 | 0.022 | [−0.169, 0.207] | 0.422 | 0.674 | 0.045 | [−0.168, 0.225] |
| 19 | allReg | IV | avg | 117 | 1.138 | 0.258 | 0.106 | [−0.076, 0.264] | 0.675 | 0.501 | 0.063 | [−0.111, 0.212] |
| 20 | allReg | IV | csd | 117 | 1.307 | 0.194 | 0.121 | [−0.096, 0.312] | −1.024 | 0.308 | −0.095 | [−0.265, 0.093] |
| 21 | allReg | V | avg | 53 | 2.489 | 0.016 | 0.329 | [0.106, 0.517] | 1.463 | 0.150 | 0.201 | [−0.023, 0.409] |
| 22 | allReg | V | csd | 53 | 0.857 | 0.395 | 0.119 | [−0.127, 0.361] | −0.220 | 0.827 | −0.031 | [−0.262, 0.210] |
| 23 | DReg | I | avg | 29 | 0.980 | 0.336 | 0.185 | [−0.212, 0.531] | 0.320 | 0.751 | 0.061 | [−0.470, 0.522] |
| 24 | DReg | I | csd | 29 | −1.303 | 0.204 | −0.243 | [−0.540, 0.091] | −0.504 | 0.618 | −0.097 | [−0.501, 0.337] |
| 25 | DReg | III | avg | 27 | 0.278 | 0.784 | 0.055 | [−0.268, 0.426] | 0.055 | 0.957 | 0.011 | [−0.273, 0.238] |
| 26 | DReg | III | csd | 27 | 0.407 | 0.688 | 0.081 | [−0.254, 0.413] | 1.347 | 0.190 | 0.260 | [−0.047, 0.498] |
| 27 | DReg | IV | avg | 22 | 1.754 | 0.095 | 0.365 | [0.012, 0.643] | −0.114 | 0.910 | −0.026 | [−0.459, 0.427] |
| 28 | DReg | IV | csd | 22 | 1.938 | 0.067 | 0.398 | [−0.043, 0.632] | −0.415 | 0.683 | −0.092 | [−0.519, 0.351] |
| 29 | DReg | V | avg | 24 | 1.497 | 0.149 | 0.304 | [0.045, 0.563] | −0.367 | 0.717 | −0.078 | [−0.444, 0.285] |
| 30 | DReg | V | csd | 24 | 0.789 | 0.438 | 0.166 | [−0.264, 0.501] | 0.635 | 0.532 | 0.134 | [−0.320, 0.615] |

**DvsHC** – diagnosed and healthy controls; **SvsHC** – sub-clinical and healthy controls; **allReg** – linear regression for all subjects together; **DReg** – linear regression for only diagnosed subjects; **avg** – average reference; **csd** – current source density.

**Table 2.** Results for all channel-pair analyses corrected for confounds.

Each row represents two channel-pair results for a given contrast, study, and space combination; uncorrected for multiple comparisons. Electrode placement for each study is shown in *Figure 1C*. (*N*: number of participants included in given contrast; *ES*: effect size; Cohen's d for group comparison and Pearson's r for regression; *CI*: bootstrap 95% confidence interval for the effect size).

| | | | | | Selected electrodes corrected for confounds | | | | | | | |
|---|---|---|---|---|---|---|---|---|---|---|---|---|
| | | | | | Pair 1 (F3–F4) | | | | Pair 2 (F7–F8) | | | |
| No. | Contrast | Study | Space | N | t | p | ES | CI | t | p | ES | CI |
| 1 | DvsHC | I | avg | 29 vs 22 | −2.679 | 0.010 | −0.789 | [−1.413,−0.218] | −0.782 | 0.438 | −0.230 | [−0.691, 0.288] |
| 2 | DvsHC | I | csd | 29 vs 22 | 0.269 | 0.789 | 0.079 | [−0.501, 0.681] | 0.305 | 0.762 | 0.090 | [−0.404, 0.622] |
| 3 | DvsHC | III | avg | 27 vs 21 | 0.204 | 0.839 | 0.064 | [−0.642, 0.684] | −1.161 | 0.252 | −0.366 | [−0.994, 0.310] |
| 4 | DvsHC | III | csd | 27 vs 21 | −0.162 | 0.872 | −0.051 | [−0.691, 0.547] | −0.703 | 0.486 | −0.221 | [−0.998, 0.558] |
| 5 | DvsHC | IV | avg | 22 vs 71 | 0.310 | 0.757 | 0.077 | [−0.338, 0.504] | 0.085 | 0.933 | 0.021 | [−0.412, 0.450] |
| 6 | DvsHC | IV | csd | 22 vs 71 | 0.978 | 0.331 | 0.244 | [−0.250, 0.816] | −1.484 | 0.141 | −0.370 | [−0.842, 0.135] |
| 7 | DvsHC | V | avg | 24 vs 29 | 1.862 | 0.069 | 0.540 | [0.033, 1.044] | 1.817 | 0.075 | 0.527 | [−0.008, 1.101] |
| 8 | DvsHC | V | csd | 24 vs 29 | 0.518 | 0.607 | 0.150 | [−0.415, 0.742] | −0.722 | 0.474 | −0.209 | [−0.689, 0.235] |
| 9 | SvsHC | II | avg | 23 vs 28 | 0.351 | 0.728 | 0.105 | [−0.444, 0.746] | −0.654 | 0.516 | −0.196 | [−0.883, 0.450] |
| 10 | SvsHC | II | csd | 23 vs 28 | 1.293 | 0.203 | 0.387 | [−0.152, 0.943] | 0.035 | 0.972 | 0.010 | [−0.653, 0.532] |
| 11 | SvsHC | III | avg | 33 vs 21 | −1.169 | 0.248 | −0.350 | [−0.946, 0.285] | −1.768 | 0.084 | −0.529 | [−1.071, 0.029] |
| 12 | SvsHC | III | csd | 33 vs 21 | −1.382 | 0.173 | −0.414 | [−1.086, 0.160] | −1.197 | 0.237 | −0.358 | [−0.997, 0.212] |
| 13 | SvsHC | IV | avg | 21 vs 71 | 1.058 | 0.293 | 0.269 | [−0.232, 0.776] | 0.466 | 0.642 | 0.118 | [−0.293, 0.515] |
| 14 | SvsHC | IV | csd | 21 vs 71 | 0.739 | 0.462 | 0.188 | [−0.302, 0.649] | −0.263 | 0.793 | −0.067 | [−0.524, 0.483] |
| 15 | allReg | I | avg | 54 | −1.352 | 0.182 | −0.188 | [−0.440, 0.078] | −0.349 | 0.728 | −0.049 | [−0.369, 0.240] |
| 16 | allReg | I | csd | 54 | −0.431 | 0.668 | −0.061 | [−0.313, 0.187] | −0.019 | 0.985 | −0.003 | [−0.335, 0.304] |
| 17 | allReg | III | avg | 91 | 0.233 | 0.816 | 0.025 | [−0.156, 0.263] | −1.170 | 0.245 | −0.127 | [−0.313, 0.092] |
| 18 | allReg | III | csd | 91 | −0.005 | 0.996 | −0.001 | [−0.190, 0.204] | −0.060 | 0.953 | −0.007 | [−0.238, 0.224] |
| 19 | allReg | IV | avg | 116 | 0.904 | 0.368 | 0.085 | [−0.088, 0.237] | 0.355 | 0.723 | 0.034 | [−0.139, 0.197] |
| 20 | allReg | IV | csd | 116 | 1.461 | 0.147 | 0.137 | [−0.064, 0.316] | −1.300 | 0.196 | −0.122 | [−0.296, 0.059] |
| 21 | allReg | V | avg | 53 | 1.685 | 0.099 | 0.236 | [−0.014, 0.449] | 1.544 | 0.129 | 0.218 | [−0.034, 0.433] |
| 22 | allReg | V | csd | 53 | 0.769 | 0.446 | 0.110 | [−0.148, 0.339] | −0.058 | 0.954 | −0.008 | [−0.214, 0.185] |
| 23 | DReg | I | avg | 29 | 0.767 | 0.450 | 0.152 | [−0.324, 0.503] | 0.265 | 0.793 | 0.053 | [−0.452, 0.564] |
| 24 | DReg | I | csd | 29 | −1.273 | 0.215 | −0.247 | [−0.588, 0.156] | −0.506 | 0.617 | −0.101 | [−0.504, 0.437] |
| 25 | DReg | III | avg | 27 | −0.054 | 0.958 | −0.012 | [−0.434, 0.485] | −0.079 | 0.937 | −0.018 | [−0.425, 0.346] |
| 26 | DReg | III | csd | 27 | 0.055 | 0.957 | 0.012 | [−0.407, 0.408] | 1.375 | 0.184 | 0.294 | [−0.096, 0.614] |
| 27 | DReg | IV | avg | 22 | 1.979 | 0.063 | 0.423 | [−0.001, 0.719] | −0.062 | 0.951 | −0.015 | [−0.409, 0.473] |
| 28 | DReg | IV | csd | 22 | 1.761 | 0.095 | 0.383 | [0.004, 0.674] | −0.269 | 0.791 | −0.063 | [−0.538, 0.418] |
| 29 | DReg | V | avg | 24 | 0.926 | 0.366 | 0.208 | [−0.209, 0.565] | −0.707 | 0.488 | −0.160 | [−0.571, 0.278] |
| 30 | DReg | V | csd | 24 | 0.723 | 0.478 | 0.164 | [−0.345, 0.554] | 0.298 | 0.769 | 0.068 | [−0.569, 0.735] |

**DvsHC** – diagnosed and healthy controls; **SvsHC** – sub-clinical and healthy controls; **allReg** – linear regression for all subjects together; **DReg** – linear regression for only diagnosed subjects; **avg** – average reference; **csd** – current source density.

test the relationship between psychometric depression score and FAA. Group contrasts included: comparison between diagnosed and healthy controls (*DvsHC*) or sub-clinical and healthy controls (*SvsHC*). The inclusion of *SvcHC* contrast is motivated by the fact that in some FAA studies depression is not diagnosed by conducting a structured clinical interview – instead groups are created based on score thresholds from psychometric depression questionnaires (for example, *De Raedt et al., 2008*; *Imperatori et al., 2019*; *Schaffer et al., 1983*). For group contrasts we used Welch t-test, which does not assume equal variance of the compared groups (*Delacre et al., 2017*). Linear contrasts were performed either for all subjects together (*allReg*) or only for the diagnosed subjects

(*DReg*). *allReg* contrast quantifies the linear relationship between FAA and BDI across all participants while *DReg* contrast tests whether FAA increases with depression severity measured with BDI questionnaire (or PHQ-9 in Study V).

Combining all the analytical pathways (*studies × statistical contrasts × analysis spaces × analysis approaches × control for confounds*) leads to 270 analyses, the results of these analyses are summarized in *Tables 1–8*.

### Channel-pair analyses

We report results of all the channel-pair analyses in *Tables 1* and *2*. Only 3 out of 60 analyses without control for confounds gave significant results. This is expected by chance, p=0.583, binomial test for a probability of significant result greater than 5%. For analyses controlling for confounds only 1 out of 60 was significant, which is also expected by chance, p=0.954, binomial test. We focus the description below on results from the analyses without control for confounds.

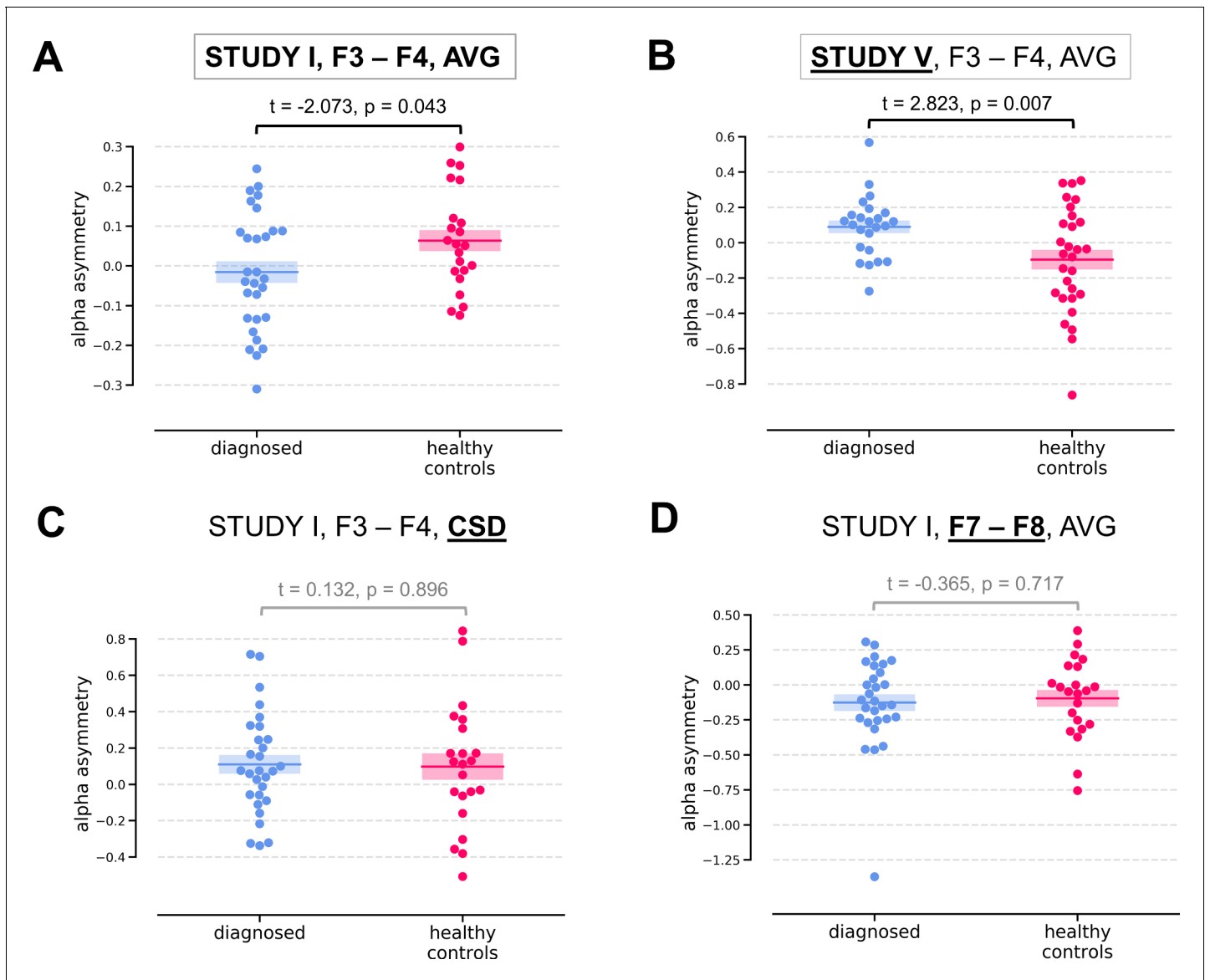

**Figure 3.** Selected results for channel-pairs level analyses, depressed vs healthy controls (*DvsHC*) contrast. Panels A and B show 2 of 3 significant channel pair results: average referenced F3–F4 channel pair for Studies I (**A**) and V (**B**). The remaining panels (**C** and **D**) show other channel pair analysis variants: specifically those that differ by exactly one parameter (underlined text) from the result presented in the panel **A**. Horizontal lines represent averages for each group, and shaded areas show standard error of the mean.

Two of the significant results were found for the diagnosed vs healthy controls contrast (*DvsHC*) for the average referenced (AVG) F3–F4 channel pair in Study I and V. In Study I right–left alpha asymmetry was lower for the diagnosed group than for healthy controls, t = −2.073, p=0.043 (*Figure 3A*). However, Study V showed the reverse: alpha asymmetry was higher for the diagnosed group compared to healthy controls, t = 2.823, p=0.007 (*Figure 3B*). Because we use right minus left alpha asymmetry only results with more negative asymmetry values in depressed individuals (negative t values) are congruent with the classical FAA effect.

**Table 3.** Results for cluster-based permutation test on frontal asymmetry space.

Each row represents cluster-based results for a given contrast, study, and space combination (*N:* number of participants included in given contrast; *min t, max t:* lowest and highest t value in the search space, respectively; *n significant points*: total number of significant points in the search space before cluster-based correction; *n clusters*: number of clusters found in given analysis; *largest cluster size:* number of channels participating in the cluster; *largest cluster p*: p-value for the largest cluster, *NA* means that no cluster was found in given analysis).

| | | | | | | | Cluster-based permutation test on frontal asymmetry space | | | |
|---|---|---|---|---|---|---|---|---|---|---|
| No. | Contrast | Study | Space | N | Min t | Max t | n significant points | n clusters | Largest cluster size | Largest cluster p |
| 1 | DvsHC | I | avg | 29 vs 22 | −2.110 | 0.023 | 3 | 1 | 3 | 0.069 |
| 2 | DvsHC | I | csd | 29 vs 22 | −0.720 | 1.983 | 0 | 0 | NA | NA |
| 3 | DvsHC | III | avg | 27 vs 21 | −1.172 | 1.058 | 0 | 0 | NA | NA |
| 4 | DvsHC | III | csd | 27 vs 21 | −1.258 | 1.875 | 0 | 0 | NA | NA |
| 5 | DvsHC | IV | avg | 22 vs 72 | −0.751 | 2.069 | 1 | 1 | 1 | 0.345 |
| 6 | DvsHC | IV | csd | 22 vs 72 | −1.600 | 1.142 | 0 | 0 | NA | NA |
| 7 | DvsHC | V | avg | 24 vs 29 | −1.260 | 2.823 | 6 | 2 | 5 | 0.026 |
| 8 | DvsHC | V | csd | 24 vs 29 | −2.901 | 1.425 | 2 | 2 | 1 | 0.156 |
| 9 | SvsHC | II | avg | 23 vs 28 | −2.581 | 0.201 | 1 | 1 | 1 | 0.164z |
| 10 | SvsHC | II | csd | 23 vs 28 | −0.855 | 1.254 | 0 | 0 | NA | NA |
| 11 | SvsHC | III | avg | 33 vs 21 | −1.410 | 1.021 | 0 | 0 | NA | NA |
| 12 | SvsHC | III | csd | 33 vs 21 | −2.315 | 2.101 | 2 | 2 | 1 | 0.227 |
| 13 | SvsHC | IV | avg | 21 vs 72 | −1.356 | 2.760 | 2 | 1 | 2 | 0.052 |
| 14 | SvsHC | IV | csd | 21 vs 72 | −1.193 | 1.296 | 0 | 0 | NA | NA |
| 15 | allReg | I | avg | 54 | −1.519 | 0.022 | 0 | 0 | NA | NA |
| 16 | allReg | I | csd | 54 | −0.727 | 1.052 | 0 | 0 | NA | NA |
| 17 | allReg | III | avg | 91 | −1.207 | 0.906 | 0 | 0 | NA | NA |
| 18 | allReg | III | csd | 91 | −1.290 | 2.210 | 1 | 1 | 1 | 0.287 |
| 19 | allReg | IV | avg | 117 | −1.807 | 2.153 | 1 | 1 | 1 | 0.202 |
| 20 | allReg | IV | csd | 117 | −1.173 | 1.353 | 0 | 0 | NA | NA |
| 21 | allReg | V | avg | 53 | −1.001 | 2.489 | 3 | 1 | 3 | 0.077 |
| 22 | allReg | V | csd | 53 | −3.291 | 1.552 | 2 | 2 | 1 | 0.187 |
| 23 | DReg | I | avg | 29 | −0.159 | 1.552 | 0 | 0 | NA | NA |
| 24 | DReg | I | csd | 29 | −1.303 | 0.487 | 0 | 0 | NA | NA |
| 25 | DReg | III | avg | 27 | −2.041 | 0.976 | 0 | 0 | NA | NA |
| 26 | DReg | III | csd | 27 | −1.420 | 1.662 | 0 | 0 | NA | NA |
| 27 | DReg | IV | avg | 22 | −1.155 | 1.754 | 0 | 0 | NA | NA |
| 28 | DReg | IV | csd | 22 | −0.415 | 1.938 | 0 | 0 | NA | NA |
| 29 | DReg | V | avg | 24 | −1.482 | 1.497 | 0 | 0 | NA | NA |
| 30 | DReg | V | csd | 24 | −2.254 | 1.554 | 1 | 1 | 1 | 0.555 |

**DvsHC** – diagnosed and healthy controls; **SvsHC** – sub-clinical and healthy controls; **allReg** – linear regression for all subjects together; **DReg** – linear regression for only diagnosed subjects; **avg** – average reference; **csd** – current source density.

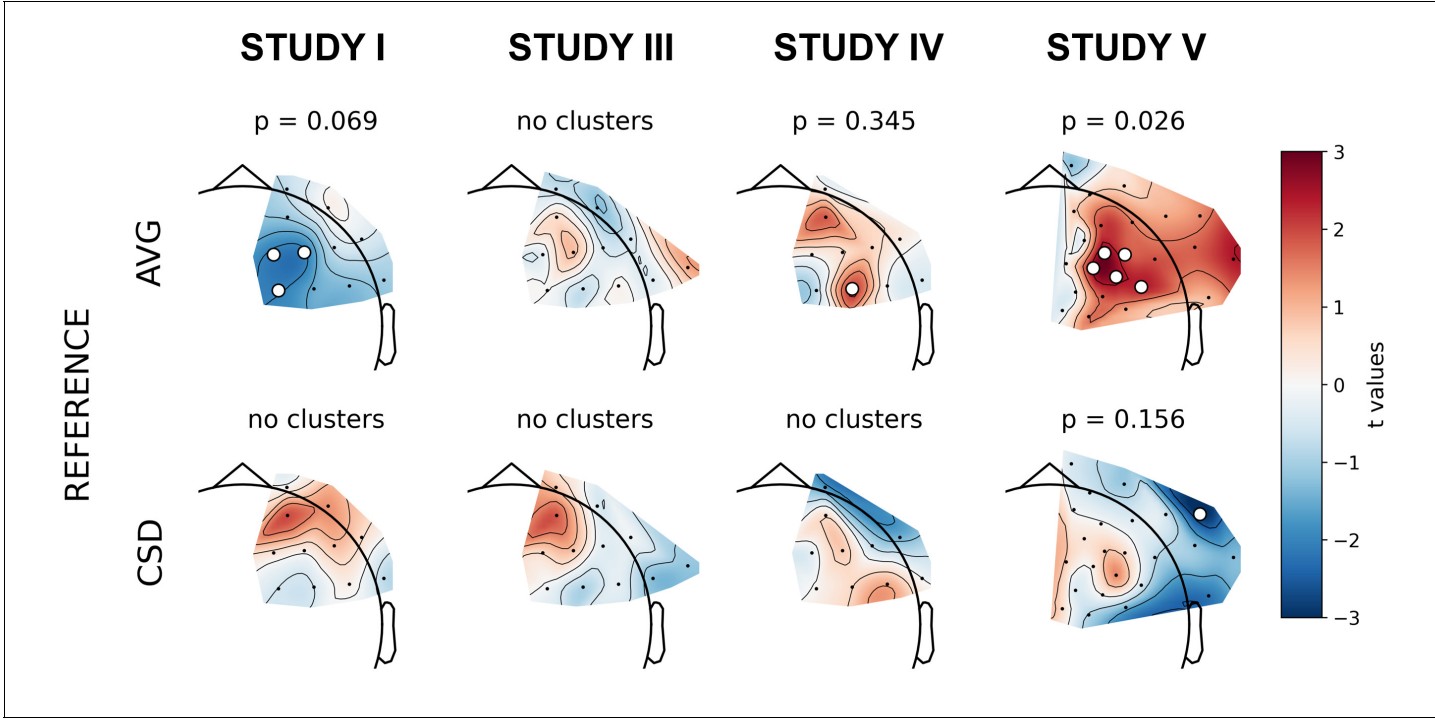

**Figure 4.** Selected results of cluster-based analyses: topographies of *DvsHC* contrast effects in a reference by study matrix. More positive (red) t values indicate more right-sided (less left-sided) alpha asymmetry for diagnosed participants. More negative (blue) t values indicate more right-sided (less left-sided) FAA for healthy controls. Channels that are part of a cluster are marked with white dots.

The online version of this article includes the following figure supplement(s) for figure 4:

**Figure supplement 1.** Results of cluster-based analyses for *allReg* contrast (regression on all subjects).

**Figure supplement 2.** Results of cluster-based analyses for *DReg* contrast (regression on diagnosed subjects).

**Figure supplement 3.** Results of cluster-based analyses for *SvsHC* (comparison between subclinical and healthy controls) contrast.

**Figure supplement 4.** An example of a more detailed visualization of asymmetry effects: group-level averages for DvsHC, Study V, AVG from F.

We now examine the FAA-congruent result from Study I in a wider context, comparing it to the outcome of related analyses that differ in a single parameter. Applying CSD instead of average reference to the same channel pair did not reveal a significant result (same study, same contrast, F3–F4 pair, CSD reference, t = 0.132, p=0.896). The results were also not significant for the other channel pair, irrespective of reference (same study, same contrast, F7–F8 pair: average reference, t = −0.365, p=0.717; CSD reference, t = 0.553, p=0.583). Performing the same analysis on data from Study III or IV also did not give rise to a significant outcome (same contrast, F3–F4 pair, AVG reference, Study III: t = 0.904, p=0.371; Study IV: t = 0.450, p=0.654). For Study II the *DvsHC* contrast was not available, but conceptually closest contrast – Sub-clinical vs Healthy Controls (*SvsHC*) – was found insignificant for both channel pairs using either AVG or CSD reference. Only this FAA-congruent *DvsHC* effect on F3–F4, AVG in Study I survives control for confounding variables (t = −2.679, p=0.010).

Another significant result was observed for linear relationship between FAA and BDI score (*allReg* contrast) on the average referenced F3–F4 channel pair in Study V: more positive values of alpha asymmetry were associated with higher BDI scores (t = 2.489, p=0.016). This result is not consistent with the standard FAA effect, which, when calculated as right minus left alpha, should manifest as a negative correlation (negative t).

However, as we argued in the introduction, single channel analyses are not a particularly good approach to testing FAA. They may be sensitive to small changes in the topography pattern and do not provide any information about the source or physiological plausibility of the effect.

## Cluster-based analyses

The next set of analyses consisted of cluster-based analyses on all frontal channel pairs (*Tables 3* and *4*). This approach gives a better view of the whole FAA space (correcting for multiple comparisons) especially when coupled with presentation of the effects' topographies. We observe only one significant effect out of 30 for standard analyses (p=0.785, binomial test) and one out of 30 (p=0.785) when controlling for confounding variables.

**Table 4.** Results for cluster-based permutation test on frontal asymmetry space corrected for confounds.

Each row represents cluster-based results for a given contrast, study and space combination (*N:* number of participants included in given contrast; *min t, max t:* lowest and highest t value in the search space, respectively; *n significant points:* total number of significant points in the search space before cluster-based correction; *n clusters:* number of clusters found in given analysis; *largest cluster size:* number of channels participating in the cluster; *largest cluster p:* p-value for the largest cluster, *NA* means that no cluster was found in given analysis).

| | | | | | Cluster-based permutation test on frontal asymmetry space corrected for confounds | | | | | |
| No. | Contrast | Study | Space | N | Min t | Max t | n significant points | n clusters | Largest cluster size | Largest cluster p |
|---|---|---|---|---|---|---|---|---|---|---|
| 1 | DvsHC | I | avg | 29 vs 22 | −2.679 | −0.149 | 5 | 1 | 5 | 0.023 |
| 2 | DvsHC | I | csd | 29 vs 22 | −1.020 | 1.591 | 0 | 0 | NA | NA |
| 3 | DvsHC | III | avg | 27 vs 21 | −1.506 | 1.361 | 0 | 0 | NA | NA |
| 4 | DvsHC | III | csd | 27 vs 21 | −1.310 | 1.378 | 0 | 0 | NA | NA |
| 5 | DvsHC | IV | avg | 22 vs 71 | −0.814 | 1.812 | 0 | 0 | NA | NA |
| 6 | DvsHC | IV | csd | 22 vs 71 | −1.812 | 0.978 | 0 | 0 | NA | NA |
| 7 | DvsHC | V | avg | 24 vs 29 | −1.618 | 2.341 | 3 | 3 | 1 | 0.327 |
| 8 | DvsHC | V | csd | 24 vs 29 | −3.017 | 0.738 | 2 | 2 | 1 | 0.205 |
| 9 | SvsHC | II | avg | 23 vs 28 | −2.622 | 0.351 | 1 | 1 | 1 | 0.177 |
| 10 | SvsHC | II | csd | 23 vs 28 | −0.838 | 1.742 | 0 | 0 | NA | NA |
| 11 | SvsHC | III | avg | 33 vs 21 | −1.768 | 1.218 | 0 | 0 | NA | NA |
| 12 | SvsHC | III | csd | 33 vs 21 | −2.090 | 1.899 | 1 | 1 | 1 | 0.358 |
| 13 | SvsHC | IV | avg | 21 vs 71 | −1.584 | 1.490 | 0 | 0 | NA | NA |
| 14 | SvsHC | IV | csd | 21 vs 71 | −1.400 | 0.739 | 0 | 0 | NA | NA |
| 15 | allReg | I | avg | 54 | −1.726 | 0.039 | 0 | 0 | NA | NA |
| 16 | allReg | I | csd | 54 | −0.816 | 0.987 | 0 | 0 | NA | NA |
| 17 | allReg | III | avg | 91 | −1.231 | 0.722 | 0 | 0 | NA | NA |
| 18 | allReg | III | csd | 91 | −1.540 | 2.119 | 1 | 1 | 1 | 0.335 |
| 19 | allReg | IV | avg | 116 | −1.804 | 1.852 | 0 | 0 | NA | NA |
| 20 | allReg | IV | csd | 116 | −1.300 | 1.461 | 0 | 0 | NA | NA |
| 21 | allReg | V | avg | 53 | −1.286 | 2.350 | 1 | 1 | 1 | 0.324 |
| 22 | allReg | V | csd | 53 | −3.541 | 0.832 | 2 | 2 | 1 | 0.173 |
| 23 | DReg | I | avg | 29 | −0.244 | 1.516 | 0 | 0 | NA | NA |
| 24 | DReg | I | csd | 29 | −1.291 | 0.565 | 0 | 0 | NA | NA |
| 25 | DReg | III | avg | 27 | −1.700 | 0.852 | 0 | 0 | NA | NA |
| 26 | DReg | III | csd | 27 | −1.311 | 1.574 | 0 | 0 | NA | NA |
| 27 | DReg | IV | avg | 22 | −1.173 | 2.728 | 1 | 1 | 1 | 0.121 |
| 28 | DReg | IV | csd | 22 | −0.269 | 2.235 | 1 | 1 | 1 | 0.260 |
| 29 | DReg | V | avg | 24 | −1.593 | 0.926 | 0 | 0 | NA | NA |
| 30 | DReg | V | csd | 24 | −2.075 | 2.586 | 2 | 2 | 1 | 0.376 |

DvsHC – diagnosed and healthy controls; SvsHC – sub-clinical and healthy controls; allReg – linear regression for all subjects together; DReg – linear regression for only diagnosed subjects; avg – average reference; csd – current source density.

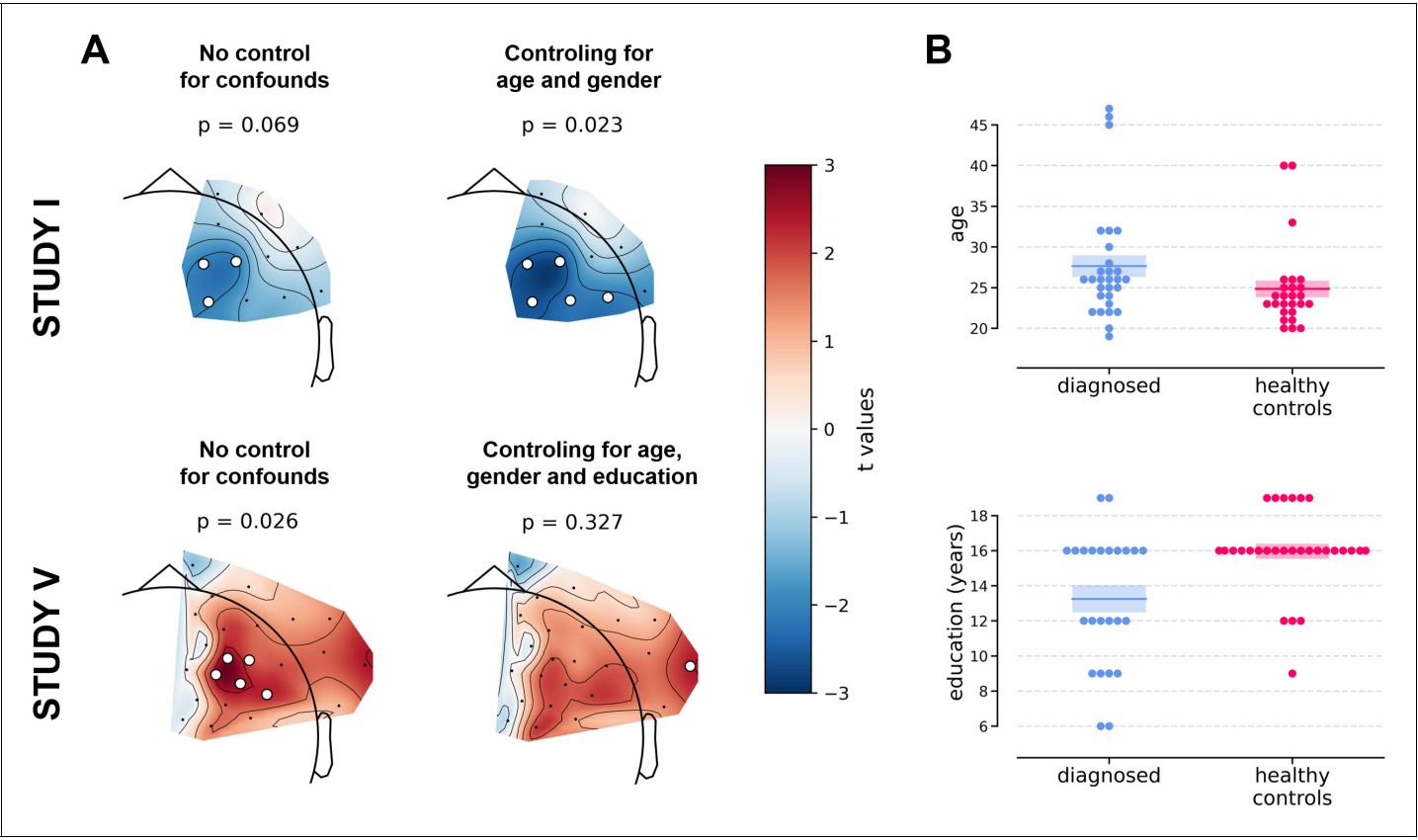

**Figure 5.** Selected results of cluster-based analyses showing the influence of statistical control for confounding variables like age, gender (Studies I and V), and education (Study V). (**A**) The logic of the topographical plots is the same as in *Figure 4*. (**B**) Swarmplots corresponding to studies in panel **A** showing the between-group difference in the selected confounding variables. Detailed results for analyses taking into account confounding variables can be found in *Table 4*.

The significant result for standard analyses corresponds to *DvsHC* contrast for the average referenced data in Study V (cluster p=0.026). This is the same analysis combination as the significant channel-pair result for Study V reported in the previous section. And just like in the previous section it represents a result that is not congruent with the classical FAA effect: the positive t values mean more positive right minus left FAA in depressed compared to healthy controls. We compare this effect to other analyses for *DvsHC* contrast in *Figure 4*. We do not observe any other significant effect in standard cluster-based results: neither contrasting sub-clinical vs healthy controls nor looking for a linear relationship between FAA and depression questionnaire score. Although single channels sometimes pass the significance threshold (see *n significant points* column in *Table 3*) these effects are not significant at the cluster level. In other words the clusters formed by these channels were not convincingly stronger from clusters observed under the null hypothesis (that is, when permuting the data).

However, we observe an interesting effect of controlling for confounds. The significant result mentioned in the previous paragraph is no longer present after controlling for age, gender, and education (see *Figure 5A*). This change might be related to the fact that the diagnosed group has significantly lower education than the control group in Study V (t = −3.07, p=0.004, see *Figure 5B*), so a part of the FAA differences between the groups can be explained away by education. On the other hand, one result close to the significance threshold in the standard analyses (*DvsHC*, Study I, AVG, cluster p=0.069) becomes significant after controlling for gender and age (p=0.023), which may be related to the difference in age between the depressed and the control group in this study (t = 2.73, p=0.0095).

## Cluster-based analyses on standardized data

All previous analyses assume that asymmetry can be detected by subtracting right and left homologous channels. Because some asymmetry effects may not match such strict left vs right pattern, we performed further analyses to alleviate this issue: in this set of analyses we relied on standardization of alpha power at frontal channels instead of right minus left subtraction. Additionally, to minimize the risk of averaging out an effect confined to a narrow frequency range, we also analyzed all

**Table 5.** Results for all cluster-based analyses on standardized data.

Each row represents cluster-based results for given contrast, study, and space (*N*: number of participants included in given contrast). Results for cluster-based permutation test on frontal asymmetry space (*min t, max t*: lowest and highest t value in the search space, respectively; *n significant points*: total number of significant points in the search space before cluster-based correction; *n clusters*: number of clusters found in given analysis; *largest cluster size:* number of channels by frequency points participating in the cluster; *largest cluster p*: p-value for the largest cluster, *NA* means that no cluster was found in given analysis).

| | | | | Cluster-based analyses on standardized data | | | | | |
|---|---|---|---|---|---|---|---|---|---|
| No. | Contrast | Study | Space | N | Min t | Max t | n significant points | n clusters | Largest cluster size | Largest cluster p |
| 1 | DvsHC | I | avg | 29 vs 22 | −1.453 | 1.434 | 0 | 0 | NA | NA |
| 2 | DvsHC | I | csd | 29 vs 22 | −2.227 | 1.879 | 3 | 2 | 2 | 0.718 |
| 3 | DvsHC | III | avg | 27 vs 21 | −1.999 | 2.326 | 2 | 1 | 2 | 0.468 |
| 4 | DvsHC | III | csd | 27 vs 21 | −2.917 | 3.882 | 23 | 3 | 16 | 0.063 |
| 5 | DvsHC | IV | avg | 22 vs 72 | −4.256 | 3.053 | 129 | 2 | 66 | 0.007 |
| 6 | DvsHC | IV | csd | 22 vs 72 | −3.180 | 2.582 | 22 | 7 | 6 | 0.259 |
| 7 | DvsHC | V | avg | 24 vs 29 | −2.516 | 2.267 | 21 | 3 | 15 | 0.223 |
| 8 | DvsHC | V | csd | 24 vs 29 | −2.703 | 2.420 | 10 | 6 | 3 | 0.723 |
| 9 | SvsHC | II | avg | 23 vs 28 | −1.685 | 2.185 | 1 | 1 | 1 | 0.774 |
| 10 | SvsHC | II | csd | 23 vs 28 | −2.059 | 1.714 | 1 | 1 | 1 | 0.941 |
| 11 | SvsHC | III | avg | 33 vs 21 | −1.906 | 1.946 | 0 | 0 | NA | NA |
| 12 | SvsHC | III | csd | 33 vs 21 | −2.284 | 3.091 | 7 | 4 | 3 | 0.577 |
| 13 | SvsHC | IV | avg | 21 vs 72 | −2.319 | 2.519 | 19 | 5 | 5 | 0.306 |
| 14 | SvsHC | IV | csd | 21 vs 72 | −2.950 | 3.130 | 25 | 4 | 11 | 0.092 |
| 15 | allReg | I | avg | 54 | −1.295 | 1.450 | 0 | 0 | NA | NA |
| 16 | allReg | I | csd | 54 | −2.082 | 1.899 | 1 | 1 | 1 | 1.000 |
| 17 | allReg | III | avg | 91 | −1.814 | 1.949 | 0 | 0 | NA | NA |
| 18 | allReg | III | csd | 91 | −2.626 | 2.547 | 14 | 4 | 6 | 0.692 |
| 19 | allReg | IV | avg | 117 | −3.406 | 2.701 | 75 | 4 | 40 | 0.079 |
| 20 | allReg | IV | csd | 117 | −2.644 | 2.918 | 32 | 5 | 13 | 0.179 |
| 21 | allReg | V | avg | 53 | −2.752 | 2.321 | 22 | 6 | 13 | 0.477 |
| 22 | allReg | V | csd | 53 | −3.733 | 2.023 | 15 | 6 | 5 | 0.891 |
| 23 | DReg | I | avg | 29 | −2.011 | 1.936 | 0 | 0 | NA | NA |
| 24 | DReg | I | csd | 29 | −2.721 | 2.093 | 8 | 5 | 3 | 1.000 |
| 25 | DReg | III | avg | 27 | −4.167 | 3.824 | 89 | 2 | 53 | 0.025 |
| 26 | DReg | III | csd | 27 | −3.525 | 3.285 | 34 | 4 | 17 | 0.105 |
| 27 | DReg | IV | avg | 22 | −1.802 | 2.213 | 1 | 1 | 1 | 0.888 |
| 28 | DReg | IV | csd | 22 | −2.219 | 2.890 | 12 | 4 | 9 | 0.316 |
| 29 | DReg | V | avg | 24 | −2.926 | 4.155 | 58 | 7 | 18 | 0.354 |
| 30 | DReg | V | csd | 24 | −3.551 | 3.084 | 46 | 9 | 14 | 0.387 |

DvsHC – diagnosed and healthy controls; **SvsHC** – sub-clinical and healthy controls; **allReg** – linear regression for all subjects together; **DReg** – linear regression for only diagnosed subjects; **avg** – average reference; csd – current source density.

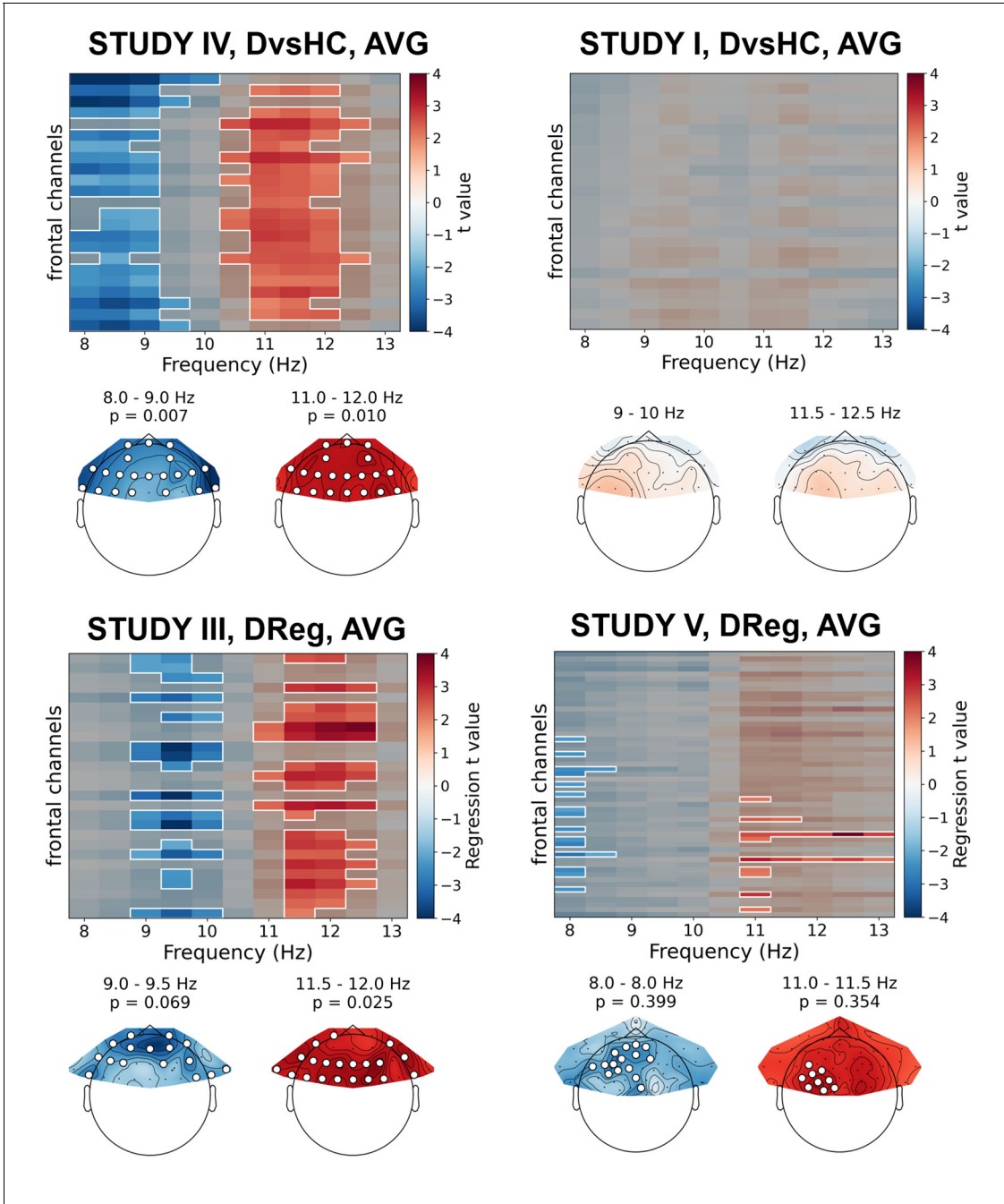

**Figure 6.** Selected results of cluster-based analyses on standardized data. Heatmaps in the upper part of each panel represent regression t values for channel by frequency search space. More positive/negative t values indicate higher/lower power with higher BDI. Clusters are indicated in the heatmaps with white outline. In each panel we present two topographies below the heatmap: showing average effect for lower and higher frequency ranges determined by the positions of the clusters. Channels that are part of a cluster are marked with white dots in the topographical plots. The online version of this article includes the following figure supplement(s) for figure 6:

**Figure supplement 1.** Results of cluster-based analyses on standardized data for *allReg* contrast (linear regression between FAA and BDI on all subjects together).

**Figure supplement 2.** Results of cluster-based analyses on standardized data for *DReg* contrast (linear regression between FAA and BDI restricted to the non-diagnosed subjects).

**Figure supplement 3.** Results of cluster-based analyses on standardized data for *DvsHC* contrast (comparison between diagnosed and healthy controls).

**Figure supplement 4.** Results of cluster-based analyses on standardized data for *SvsHC* contrast (comparison between subclinical and healthy controls).

**Table 6.** Results for all cluster-based analyses on standardized data corrected for confounds.

Each row represents cluster-based results for given contrast, study, and space (*N*: number of participants included in given contrast). Results for cluster-based permutation test on frontal asymmetry space (*min t, max t*: lowest and highest t value in the search space, respectively; *n significant points*: total number of significant points in the search space before cluster-based correction; *n clusters*: number of clusters found in given analysis; *largest cluster size:* number of channels by frequency points participating in the cluster; *largest cluster p*: p-value for the largest cluster, *NA* means that no cluster was found in given analysis).

| | | | | | Cluster-based analyses on standardized data corrected for confounds | | | | | |
| No. | Contrast | Study | Space | N | Min t | Max t | n significant points | n clusters | Largest cluster size | Largest cluster p |
|---|---|---|---|---|---|---|---|---|---|---|
| 1 | DvsHC | I | avg | 29 vs 22 | −1.766 | 1.716 | 0 | 0 | NA | NA |
| 2 | DvsHC | I | csd | 29 vs 22 | −2.007 | 1.614 | 0 | 0 | NA | NA |
| 3 | DvsHC | III | avg | 27 vs 21 | −1.387 | 1.896 | 0 | 0 | NA | NA |
| 4 | DvsHC | III | csd | 27 vs 21 | −3.345 | 3.054 | 17 | 4 | 6 | 0.567 |
| 5 | DvsHC | IV | avg | 22 vs 71 | −4.295 | 3.304 | 143 | 2 | 72 | 0.006 |
| 6 | DvsHC | IV | csd | 22 vs 71 | −3.095 | 2.854 | 33 | 5 | 12 | 0.179 |
| 7 | DvsHC | V | avg | 24 vs 29 | −1.982 | 1.976 | 0 | 0 | NA | NA |
| 8 | DvsHC | V | csd | 24 vs 29 | −2.026 | 2.798 | 10 | 4 | 7 | 0.772 |
| 9 | SvsHC | II | avg | 23 vs 28 | −1.511 | 1.841 | 0 | 0 | NA | NA |
| 10 | SvsHC | II | csd | 23 vs 28 | −2.059 | 1.876 | 1 | 1 | 1 | 1.000 |
| 11 | SvsHC | III | avg | 33 vs 21 | −2.964 | 2.422 | 22 | 5 | 10 | 0.374 |
| 12 | SvsHC | III | csd | 33 vs 21 | −2.424 | 3.562 | 12 | 4 | 4 | 0.865 |
| 13 | SvsHC | IV | avg | 21 vs 71 | −2.667 | 2.858 | 34 | 3 | 31 | 0.113 |
| 14 | SvsHC | IV | csd | 21 vs 71 | −2.389 | 3.317 | 23 | 3 | 10 | 0.273 |
| 15 | allReg | I | avg | 54 | −1.404 | 1.614 | 0 | 0 | NA | NA |
| 16 | allReg | I | csd | 54 | −1.752 | 2.155 | 4 | 2 | 2 | 1.000 |
| 17 | allReg | III | avg | 91 | −2.119 | 1.963 | 5 | 1 | 5 | 0.612 |
| 18 | allReg | III | csd | 91 | −2.558 | 2.825 | 25 | 3 | 17 | 0.112 |
| 19 | allReg | IV | avg | 116 | −3.414 | 3.009 | 92 | 4 | 54 | 0.038 |
| 20 | allReg | IV | csd | 116 | −2.571 | 3.112 | 32 | 6 | 11 | 0.216 |
| 21 | allReg | V | avg | 53 | −1.942 | 2.057 | 1 | 1 | 1 | 1.000 |
| 22 | allReg | V | csd | 53 | −3.005 | 2.362 | 11 | 5 | 5 | 0.945 |
| 23 | DReg | I | avg | 29 | −1.902 | 1.990 | 0 | 0 | NA | NA |
| 24 | DReg | I | csd | 29 | −2.583 | 2.088 | 9 | 5 | 4 | 0.955 |
| 25 | DReg | III | avg | 27 | −3.938 | 3.530 | 70 | 4 | 36 | 0.073 |
| 26 | DReg | III | csd | 27 | −3.198 | 4.098 | 39 | 7 | 21 | 0.035 |
| 27 | DReg | IV | avg | 22 | −2.213 | 2.662 | 4 | 2 | 2 | 0.659 |
| 28 | DReg | IV | csd | 22 | −3.056 | 3.114 | 16 | 4 | 10 | 0.265 |
| 29 | DReg | V | avg | 24 | −3.200 | 4.246 | 84 | 7 | 27 | 0.266 |
| 30 | DReg | V | csd | 24 | −3.533 | 3.378 | 78 | 7 | 36 | 0.092 |

**DvsHC** – diagnosed and healthy controls; **SvsHC** – sub-clinical and healthy controls; **allReg** – linear regression for all subjects together; **DReg** – linear regression for only diagnosed subjects; **avg** – average reference; **csd** – current source density.

frequency bins in the 8–13 Hz range. This strategy is used in a set of 60 analyses (30 without and 30 with control for confounds) – we report their results in *Tables 5* and *6*.

Only two out of 30 analyses on standardized data without control for confounds showed statistically significant result – this is expected by chance given our alpha level (p=0.447, binomial test). Specifically, the significant results were found for: (a) *DReg* contrast, Study III (cluster p=0.025) and (b) *DvsHC* contrast, Study IV (p=0.007), both for average referenced data. Both these effects are potentially interesting, because they show a similar pattern – a positive relationship between (a)

depression severity (BDI score, only the diagnosed subjects) or (b) diagnosis status and power in the higher alpha band (*DReg*: 11–12.5 Hz, *DvsHC*: 10.5–12 Hz) across many frontal channels (see *Figure 6*). Because it is accompanied with an inverse effect in a lower frequency band (*DReg*: 9–10 Hz, *DvsHC*: 8–9 Hz) at similar channels, averaging across the whole 8–13 Hz frequency range could lead to both effects cancelling each other in the average. Such a pattern of inverse effects across frequencies could arise due to a frequency shift of the individual alpha peak or the narrowing of the peak with depression severity but it may also be a direct consequence of standardization. However, both effects do not represent FAA as their topography is symmetrical (*DReg*: $\chi^2(1)=0.153$, p=0.696, *DvsHC*: $\chi^2(1)=0.000$, p=1) and were not replicated when using CSD reference in the same studies (Study III, *Dreg*: cluster p=0.259; Study IV, *DvsHC*: p=0.105) or performing the same contrast on data from other studies (see more: *Table 5*, *Table 6*, and *Figure 6*).

Although the conclusions that can be drawn from these standardization analyses are not in favor of FAA–DD relationship, they demonstrate the strength of the proposed approach in detecting effects that might be otherwise missed when averaging across the whole alpha range or when testing only the differences on corresponding right–left channel pairs.

## Source level analyses

Because observing the effect in the signal recorded from frontal channels does not guarantee that the source of this effect is frontal we conducted a second set of additional analyses using source localization with DICS beamforming. In these analyses the FAA was evaluated in the source space by subtracting power of the corresponding right and left hemisphere vertices.

Results of source level analyses are reported in *Tables 7* and *8*. One of the 15 source space analyses turned out statistically significant – this is expected by chance given our alpha value (p=0.537, binomial test). The same number of significant results was found when controlling for confounding variables (1/15, p=0.537).

**Table 7.** Results for all source level analyses.

Each row represents source level results for given contrast, study, and space (*N*: number of participants included in given contrast). Results for cluster-based permutation test on frontal asymmetry source space (*min t, max t*: lowest and highest t value in the search space, respectively; *n significant points*: total number of significant points in the search space before cluster-based correction; *n clusters*: number of clusters found in given analysis; *largest cluster p*: p-value for the largest cluster, *NA* means that no cluster was found in given analysis).

| No. | Contrast | Study | N | Source level analysis | | | | | |
| | | | | Min t | Max t | n significant points | n clusters | Largest cluster size | Largest cluster p |
|---|---|---|---|---|---|---|---|---|---|
| 1 | DvsHC | I | 29 vs 22 | −1.906 | 2.404 | 2 | 1 | 2 | 0.489 |
| 2 | DvsHC | III | 27 vs 21 | −2.557 | 1.158 | 29 | 4 | 14 | 0.267 |
| 3 | DvsHC | IV | 22 vs 72 | −3.921 | 1.151 | 320 | 2 | 316 | 0.010 |
| 4 | DvsHC | V | 24 vs 29 | −2.612 | 0.943 | 24 | 1 | 24 | 0.207 |
| 5 | SvsHC | II | 23 vs 28 | −0.909 | 1.321 | 0 | 0 | NA | NA |
| 6 | SvsHC | III | 34 vs 21 | −1.809 | 1.341 | 0 | 0 | NA | NA |
| 7 | SvsHC | IV | 21 vs 72 | −2.339 | 0.679 | 11 | 1 | 11 | 0.340 |
| 8 | allReg | I | 54 | −2.319 | 1.549 | 20 | 1 | 20 | 0.367 |
| 9 | allReg | III | 92 | −2.232 | 0.290 | 21 | 2 | 20 | 0.343 |
| 10 | allReg | IV | 117 | −2.220 | 1.211 | 10 | 1 | 10 | 0.466 |
| 11 | allReg | V | 53 | −2.644 | 1.201 | 15 | 3 | 6 | 0.574 |
| 12 | DReg | I | 29 | −2.292 | 1.150 | 41 | 2 | 22 | 0.328 |
| 13 | DReg | III | 27 | −2.624 | −0.375 | 26 | 3 | 18 | 0.335 |
| 14 | DReg | IV | 22 | −1.216 | 2.064 | 0 | 0 | NA | NA |
| 15 | DReg | V | 24 | −2.068 | 1.758 | 0 | 0 | NA | NA |

**DvsHC** – diagnosed and healthy controls; **SvsHC** – sub-clinical and healthy controls; **allReg** – linear regression for all subjects together; **DReg** – linear regression for only diagnosed subjects.

The significant effect was found for *DvsHC* contrast in Study IV (p=0.010, see *Figure 7*) and remains significant when controlling for confounds (cluster p=0.011; see *Table 8*). It represents more negative FAA values for depressed compared to healthy individuals, which is congruent with the traditional FAA effect.

In most of the analyses the pattern and sign of the t values points towards a more left-sided effect. For example in the *allReg* contrast: the negative t values suggest lower R–L differences in high than in low BDI participants, which means more left-sided alpha power with higher BDI. Although this pattern seems to be in line with FAA–DD literature, almost all of the source space effects are weak and do not survive the correction for multiple comparisons. However, it is important to remember that individual MRI scans and channel locations were not available in the present study: their availability would lead to lower error in source reconstruction.

## Analyses on aggregated data

Finally, to overcome the relatively low statistical power of analyses on separate data sets we aggregate data from all studies that include identical contrasts and perform analyses on the aggregated data. Before aggregation we tested whether the FAA values from both selected channel pairs have similar scale across the five studies with a Levene test. Because the scale was significantly different across studies (F3–F4: W = 8.68, p<0.0001; F7–F8: W = 5.21, p=0.002) and because such scale differences can arise from lab-specific equipment or adopted impedance threshold, we z-scored the FAA values within each study before aggregation. All aggregated channel pair analyses can be seen in *Figure 8*, *Figure 8—figure supplement 1*, and *Table 9*. For brevity we discuss only the results for *DvsC* and *allReg* contrasts for average referenced channel pairs.

Aggregated *DvsHC* contrast analysis encompasses 246 participants (102 diagnosed and 144 healthy controls) and 245 when controlling for confounds (one participant from the control group was removed due to missing information on confounding variables). For the F3–F4 channel pair the

**Table 8.** Results for all source level analyses corrected for confounds.

Each row represents source level results for given contrast, study, and space (*N*: number of participants included in given contrast) Results for cluster-based permutation test on frontal asymmetry source space (*min t, max t*: lowest and highest t value in the search space, respectively; *n significant points*: total number of significant points in the search space before cluster-based correction; *n clusters*: number of clusters found in given analysis; *largest cluster p*: p-value for the largest cluster, *NA* means that no cluster was found in given analysis).

| No. | Contrast | Study | N | Min t | Max t | n significant points | n clusters | Largest cluster size | Largest cluster p |
|---|---|---|---|---|---|---|---|---|---|
| | | | | | | Source level analysis corrected for confounds | | | |
| 1 | DvsHC | I | 29 vs 22 | −1.416 | 2.057 | 1 | 1 | 1 | 0.708 |
| 2 | DvsHC | III | 27 vs 21 | −2.161 | 1.704 | 3 | 2 | 2 | 0.555 |
| 3 | DvsHC | IV | 22 vs 71 | −3.685 | 1.066 | 365 | 2 | 346 | 0.011 |
| 4 | DvsHC | V | 24 vs 29 | −2.630 | 0.287 | 64 | 3 | 46 | 0.231 |
| 5 | SvsHC | II | 23 vs 28 | −0.814 | 1.911 | 0 | 0 | NA | NA |
| 6 | SvsHC | III | 34 vs 21 | −1.770 | 1.526 | 0 | 0 | NA | NA |
| 7 | SvsHC | IV | 21 vs 71 | −2.732 | −0.083 | 58 | 1 | 58 | 0.192 |
| 8 | allReg | I | 54 | −1.846 | 1.143 | 0 | 0 | NA | NA |
| 9 | allReg | III | 92 | −2.195 | 0.698 | 15 | 1 | 15 | 0.391 |
| 10 | allReg | IV | 116 | −2.313 | 0.910 | 44 | 4 | 19 | 0.375 |
| 11 | allReg | V | 53 | −2.901 | 0.317 | 79 | 4 | 36 | 0.264 |
| 12 | DReg | I | 29 | −2.164 | 1.010 | 15 | 2 | 11 | 0.428 |
| 13 | DReg | III | 27 | −2.991 | 0.175 | 66 | 3 | 63 | 0.186 |
| 14 | DReg | IV | 22 | −1.397 | 1.818 | 0 | 0 | NA | NA |
| 15 | DReg | V | 24 | −2.893 | 1.784 | 22 | 1 | 22 | 0.352 |

**DvsHC** – diagnosed and healthy controls; **SvsHC** – sub-clinical and healthy controls; **allReg** – linear regression for all subjects together; **DReg** – linear regression for only diagnosed subjects; **nonDReg** – linear regression for only the non-diagnosed subjects.

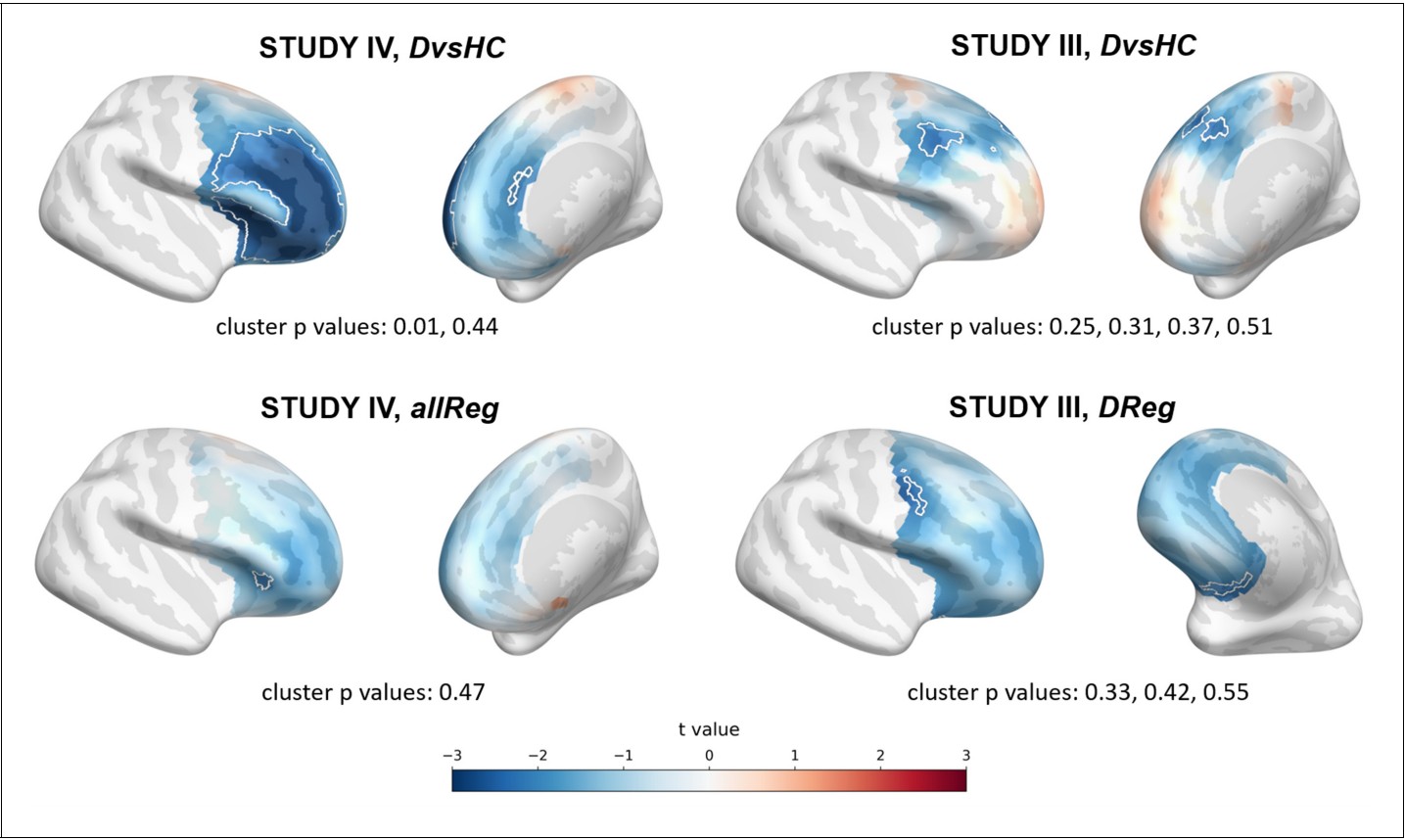

**Figure 7.** Selected results of source level analyses showing spatial t value maps for respective contrasts. Cluster limits are marked with white outlines, and corresponding cluster p-values are shown below each panel. Color bar at the bottom presents color coding for the t values.

The online version of this article includes the following figure supplement(s) for figure 7:

**Figure supplement 1.** Results of source level analyses for *allReg* contrast (linear regression between FAA and BDI on all subjects) showing spatial t value maps for regression analyses.

**Figure supplement 2.** Results of source level analyses for *DReg* contrast (linear regression between FAA and BDI restricted to diagnosed subjects) showing spatial t value maps for regression analyses.

**Figure supplement 3.** Results of source level analyses for *DvsHC* contrast (comparison between diagnosed and healthy controls) showing spatial t value maps for regression analyses.

**Figure supplement 4.** Results of source level analyses for *SvsHC* contrast (comparison between subclinical and healthy controls) showing spatial t value maps for regression analyses.

Cohen's d is 0.147 (CI = [−0.111, 0.396]) and −0.011 when controlling for confounds (CI = [−0.264, 0.244]). Both confidence intervals exclude all but small effect sizes (classical FAA effect for right–left, diagnosed – controls should be negative). The effect sizes for F7–F8 channel pair are similar: d = 0.098, CI = [−0.161, 0.338] without control for confounds and d = −0.006, CI = [−0.266, 0.240] when controlling for confounding variables.

To quantify the support for the null hypothesis we calculate Bayes factors for the null (BF01). For F3–F4 channel pair the BF01 equals 3.831, which means that the data are almost four times more likely under the null than alternative hypothesis. When controlling for confounding variables the BF01 increases to 7.042. For F7–F8 channel pair the BF01 are: 5.405 and 7.042 when controlling for confounds. Bayes factors between 3 and 10 are considered moderate evidence, so the results provide moderate evidence for no FAA difference between diagnosed and healthy individuals.

Aggregated allReg contrast analysis includes 315 participants (314 when controlling for confounds). For F3–F4 channel pair the Pearson's r is 0.085 (CI = [−0.017, 0.184]) and decreases to 0.041 when controlling for confounds (CI = [−0.063, 0.143]). Corresponding Bayes factors for the null are: 4.651 and 10.87 which suggests moderate to strong evidence for no relationship between FAA and depression score.

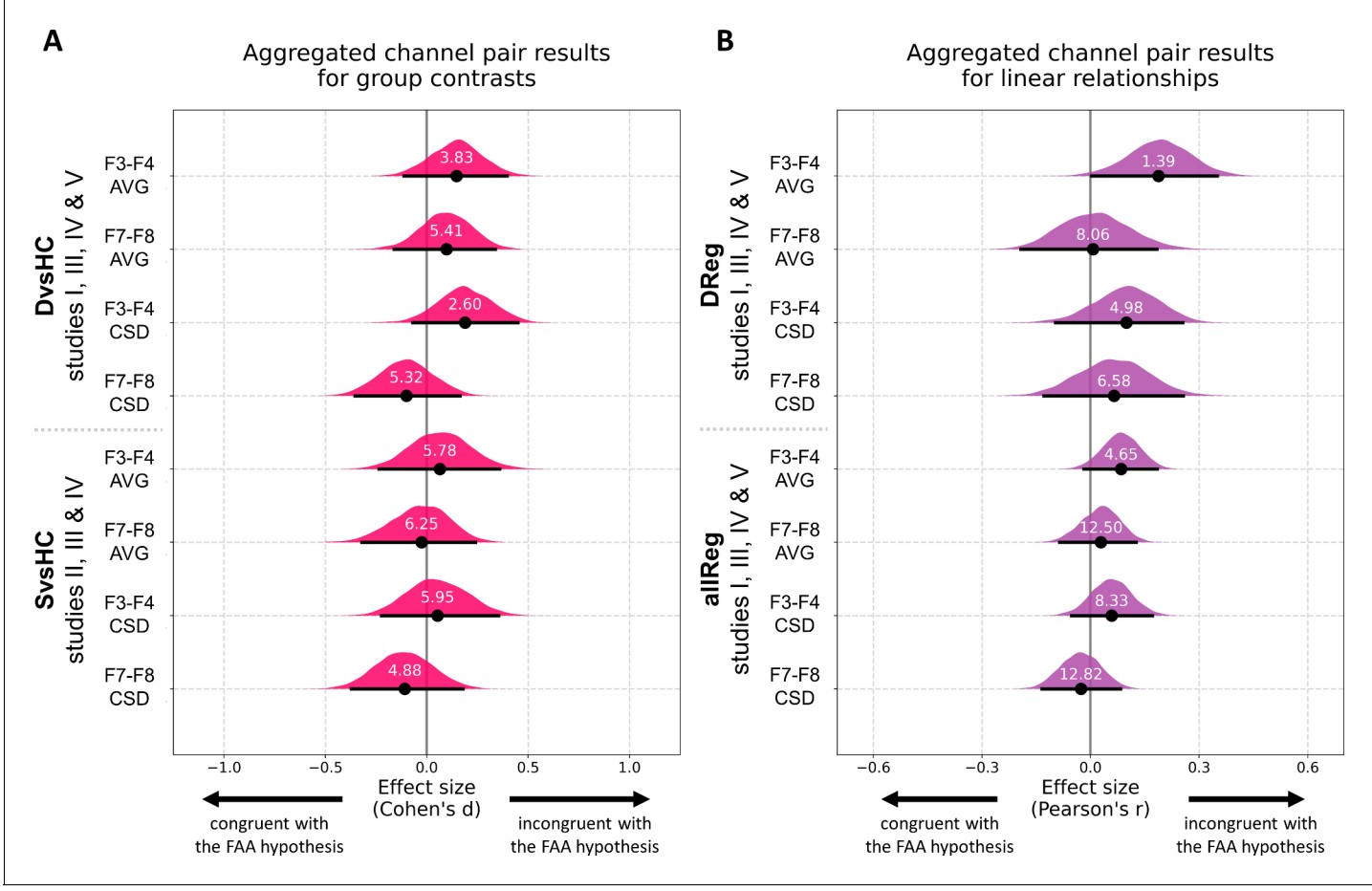

**Figure 8.** Results for channel pair analyses where studies including identical group contrasts (A) and linear contrasts (B) are combined. Each row corresponds to one analysis on a single channel pair. The contrasts, studies, and channel pairs are labeled on the y axis. The black dots correspond to observed effect sizes in Cohen's d/Pearson's r, while the black lines indicate 95% confidence intervals for the effect size estimated using bias-corrected accelerated bootstrapping. The magenta/purple shapes represent bootstrap distributions and the white numbers printed on the distributions are Bayes factors for the null hypothesis (BF01). BF01 of 4 indicates that the data are four times more likely under the null than the alternative hypothesis. BF01 between 3 and 10 are considered moderate evidence for the null hypothesis.

The online version of this article includes the following figure supplement(s) for figure 8:

**Figure supplement 1.** Results for channel pair analyses with control for confounds where studies including identical group contrasts (A) and linear contrasts (B) are combined.

**Figure supplement 2.** Results for channel pair gender × contrast interaction analyses on aggregated data with control for confounds.

We also aggregate the data from relevant studies in the source space and then perform cluster-based permutation tests for all defined contrasts. Just like in the aggregated channel pair analyses, before aggregation the data are z-scored within each study (each source vertex is z-scored across participants separately) to avoid creating non-normal distributions by joining data of different scale. The results of the aggregated source space analyses can be found in *Table 10*.

Although no analysis yields nominally significant results it is interesting to note that contrasts *DvsHC* and *allReg* are at the conventional 'trend' level (*DvsHC*: 0.077 and 0.066 when controlling for confounds; *allReg*: 0.067 and 0.054 when controlling for confounds) and their direction agrees with the traditional FAA effect.

## Discussion

We conducted a multiverse analysis of EEG data sets from five independent studies, with 388 participants and 270 analyses in total, to test the robustness and credibility of the relationship between

FAA and depressive mood. We performed 120 replicatory single channel pairs analyses and 60 corresponding cluster-based analyses. We have also conducted 90 additional analyses addressing some of the limitations of current FAA studies. Out of 270 performed analyses only 13 produced statistically significant results – a result expected by chance when using 0.05 alpha level (binomial test, p=0.595). Moreover, more than half of the significant results (8/13) are incongruent with the traditional FAA effect: by either showing the opposite direction of the effect (6) or not showing asymmetry (two significant effects on standardized data). Overall, the conducted analyses do not provide a basis to reject the null hypothesis of no relationship between resting state FAA and DD.

Our conclusion is similar to this formulated by other research groups (*Kaiser et al., 2018b*; *van der Vinne et al., 2017*), stating that treating FAA as a biomarker of DD is not sufficiently empirically grounded. As our data shows, this skepticism is not limited to single channel pair analyses – improving on the limitations of methods commonly used in FAA literature does not change the pattern of results.

Despite this, FAA is one of the most common indicators of DD with a long history of successful studies – it might be difficult to believe that all previous FAA research represents Type I errors. Therefore it is worth considering that we just fail to detect this effect here. First, the FAA effect size may be too small to be reliably observed with a small to moderate sample size. The average number of participants per study is 78 in our case, but most analyses contain around 25 participants per group, which grants sufficient power to detect mostly large effects. Although single analyses reported here can be deemed inconclusive, the whole multiverse set of analyses is incompatible with the presence of moderate to strong relationship between FAA and DD. To strengthen this point we performed analyses on data aggregated across studies (see section *Analyses on aggregated data*) showing estimated effect sizes, their confidence intervals, and Bayes factors for the null hypothesis (see *Figure 8* and *Figure 8—figure supplement 1*). All confidence intervals for diagnosed vs healthy controls group contrast (*DvsHC*) and linear relationship between FAA and diagnosis score (*allReg*) exclude strong and moderate effects in the direction compatible with traditional FAA effect. Moreover, Bayes factors indicate that there is moderate evidence for the null hypothesis (or moderate to strong evidence for the null in allReg contrast). Although we cannot exclude a small FAA–DD relationship, if the effect was in this range then most published studies would have been underpowered to detect it. This line of thought is also supported in the meta-analysis by *van der Vinne et al., 2017*, which shows that studies with larger samples were less likely to report high effect sizes. For example, the largest EEG FAA study on a sample of 1008 DD patients and 336 controls did not confirm the diagnostic value of FAA in DD (*Arns et al., 2016*). Such pattern of results suggests publication bias or that the FAA–DD effect, if it exists, is detectable only in highly selected samples and is of small magnitude on the population level.

Although the collected studies contain data from sub-clinical, mild, as well as major depression patients, we do not think our results can be explained by the level of depression severity. Clinical participants in Studies I, III, IV, and V and sub-clinical participants in Studies II and III manifested a wide range of DD symptoms indicated by BDI/PHQ-9 scores (see *Figure 1* for BDI histograms), but regression analysis (*DReg*) did not reveal any relationship between FAA and BDI score in the aggregated analyses. This means that participants with stronger DD symptoms were not better characterized by a specific FAA pattern even when we controlled for confounding variables like age, gender, and education. However, given that 56 out of 102 participants (54.9%) in the aggregated clinical group were diagnosed with mild DD, it would be interesting to repeat the analyses presented here on a data set with more major DD patients.

*Smith et al., 2017* previously suggested that the relationship between FAA and DD is stronger when the participant is given some emotion-related task, as opposed to resting condition, where the task is unspecified. Although this is possible, we wanted to stay true to the design of most FAA–DD studies, which measure EEG during rest. An interesting approach for future studies would be to compare rest blocks separated by an emotional task (see, for example, *Beeney et al., 2014*).

CSD has been previously recommended in the literature (*Kayser and Tenke, 2015*; *Stewart et al., 2014*) for studies on FAA because it reduces volume conduction and makes topographies more focal. We do not see support for the claim that CSD is more sensitive to FAA in our results. We even think it may be the contrary – almost all significant or trend-level effects disappear with CSD reference. The fact that CSD produces more focal topographies coupled with potential high variability of topographies across subjects may result in lower probability of detecting an effect.

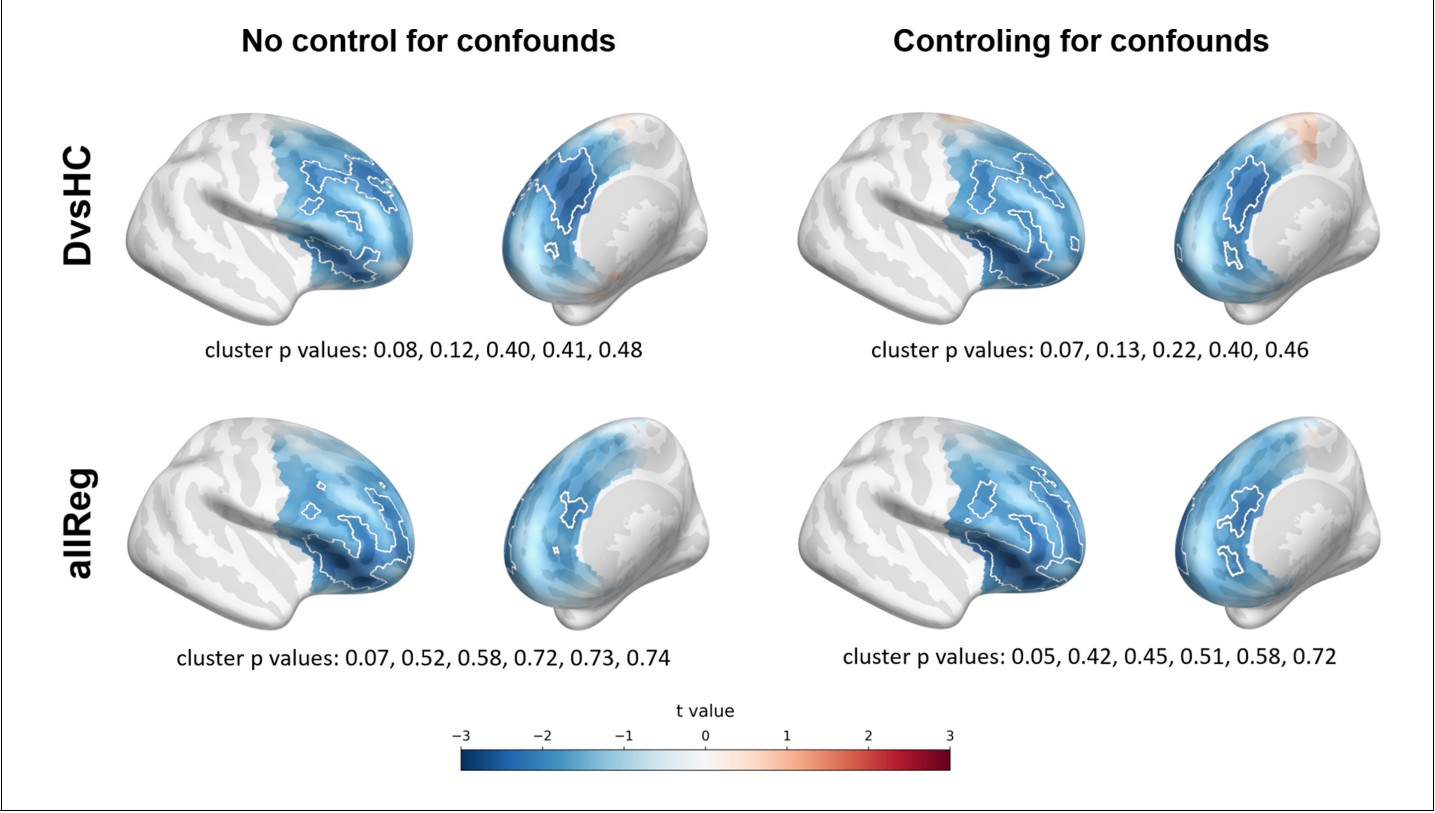

**No control for confounds** — **Controling for confounds**

DvsHC
cluster p values: 0.08, 0.12, 0.40, 0.41, 0.48

cluster p values: 0.07, 0.13, 0.22, 0.40, 0.46

allReg
cluster p values: 0.07, 0.52, 0.58, 0.72, 0.73, 0.74

cluster p values: 0.05, 0.42, 0.45, 0.51, 0.58, 0.72

t value
−3 −2 −1 0 1 2 3

**Figure 9.** Selected results for aggregated source space analyses showing spatial t value maps for respective contrasts. Cluster limits are marked with white outlines, and corresponding cluster p-values are shown below each panel. Color bar at the bottom presents color coding for the t values. The online version of this article includes the following figure supplement(s) for figure 9:

**Figure supplement 1.** Selected results for aggregated source space analyses showing spatial t value maps for respective contrasts.

**Figure supplement 2.** Results for interaction analyses (gender × contrast) for aggregated studies in source space.

Given that the orientation of the sources responsible for the effect of interest may be variable across subjects, increasing the focality of their projections may only exacerbate the issue.

In contrast to CSD, we see some indication of the traditional FAA effect in the source space analyses. Although only 2/30 source space analyses are significant, partitioning the effects into contrasts (see *Table 11*) reveals 2/8 significant *DvsHC* results, which is only barely consistent with 5% error rate (p=0.057, binomial test). Also the direction of the effect in almost all source results is consistent with the traditional FAA effect (negative t values). Finally, the same consistency of the effect direction can be observed in the source space analyses on aggregated data (see *Table 10*, *Figure 9*, and *Figure 9—figure supplement 1*) – all source space vertices with an uncorrected significant effect show negative effect direction. The *DvsHC* and *allReg* contrasts on the aggregated data are at the trend level (p=0.077 and p=0.067, respectively) and it might seem that using more rigorous source localization (individual MRI scans and channel positions) could lead to a significant effect. Nevertheless it is difficult to speculate based on statistical tendency – with more subjects or more precise source localization the results might equally likely land further away from the alpha threshold.

Previous studies have suggested that FAA effect may manifest differently depending on gender (*Jaworska et al., 2012*; *Stewart et al., 2010*; *van der Vinne et al., 2017*). Because all the analyses with control for confounding variables that we have conducted control only for the main effect of gender, we additionally tested the presence of an interaction between gender and diagnosis (or depression score) in predicting FAA. This set of analyses were restricted only to the aggregated data with control for confounds. None of these analyses demonstrated a significant interaction effect (see *Figure 8—figure supplement 2* and *Figure 9—figure supplement 2*).

**Table 9.** Results for analyses on data aggregated across studies: tests on frontal asymmetry on selected channel pairs.

Each row represents a given contrast × reference × control for confounds combination. *N*: number of participants included in given contrast, *control for confounds*: whether the FAA data was residualized with respect to confounding variables (age, gender and education); ES: effect size, measured as Cohen's d for DvsHC and SvsHC contrasts and Spearman's r for Dreg and allReg contrasts; CI: bootstrap confidence interval for the effect size; BF01: Bayes Factor for the null hypothesis.

| | | | | | Aggregated channel pair analyses | | | | | |
| | | | | | Pair 1 (F3–F4) | | | Pair 2 (F7–F8) | | |
| No. | Contrast | Space | N | Control for confounds | ES | CI | BF01 | ES | CI | BF01 |
|---|---|---|---|---|---|---|---|---|---|---|
| 1 | DvsHC | avg | 102 vs 144 | − | 0.147 | [−0.111, 0.396] | 3.831 | 0.098 | [−0.161, 0.338] | 5.405 |
| 2 | DvsHC | avg | 102 vs 143 | + | −0.011 | [−0.264, 0.244] | 7.042 | −0.006 | [−0.266, 0.240] | 7.042 |
| 3 | DvsHC | csd | 102 vs 144 | − | 0.188 | [−0.069, 0.449] | 2.597 | −0.100 | [−0.354, 0.164] | 5.319 |
| 4 | DvsHC | csd | 102 vs 143 | + | 0.103 | [−0.159, 0.362] | 5.236 | −0.164 | [−0.417, 0.108] | 3.300 |
| 5 | SvsHC | avg | 77 vs 121 | − | 0.065 | [−0.236, 0.359] | 5.780 | −0.025 | [−0.320, 0.240] | 6.250 |
| 6 | SvsHC | avg | 77 vs 120 | + | 0.024 | [−0.269, 0.315] | 6.211 | −0.144 | [−0.447, 0.140] | 4.016 |
| 7 | SvsHC | csd | 77 vs 121 | − | 0.053 | [−0.223, 0.355] | 5.952 | −0.108 | [−0.372, 0.179] | 4.878 |
| 8 | SvsHC | csd | 77 vs 120 | + | 0.053 | [−0.222, 0.350] | 5.952 | −0.110 | [−0.389, 0.179] | 4.831 |
| 9 | DReg | avg | 102 | − | 0.188 | [0.006, 0.354] | 1.387 | 0.007 | [−0.186, 0.188] | 8.065 |
| 10 | DReg | avg | 102 | + | 0.175 | [−0.018, 0.348] | 1.742 | −0.029 | [−0.211, 0.155] | 7.752 |
| 11 | DReg | csd | 102 | − | 0.099 | [−0.072, 0.260] | 4.975 | 0.065 | [−0.136, 0.252] | 6.579 |
| 12 | DReg | csd | 102 | + | 0.051 | [−0.131, 0.216] | 7.092 | 0.048 | [−0.157, 0.255] | 7.194 |
| 13 | allReg | avg | 315 | − | 0.085 | [−0.017, 0.184] | 4.651 | 0.029 | [−0.080, 0.126] | 12.5 |
| 14 | allReg | avg | 314 | + | 0.041 | [−0.063, 0.143] | 10.87 | 0.001 | [−0.120, 0.104] | 14.085 |
| 15 | allReg | csd | 315 | − | 0.059 | [−0.054, 0.166] | 8.333 | −0.026 | [−0.141, 0.082] | 12.821 |
| 16 | allReg | csd | 314 | + | 0.055 | [−0.056, 0.168] | 8.772 | −0.048 | [−0.158, 0.059] | 9.901 |

DvsHC – diagnosed and healthy controls; **SvsHC** – sub-clinical and healthy controls; **allReg** – linear regression for all subjects together; **DReg** – linear regression for only diagnosed subjects; **avg** – average reference; **csd** – current source density.

A challenge for future FAA studies would be to move beyond a 'marker-only' approach, describe the theoretical assumptions behind FAA in more detail, and let these assumptions dictate an adequate analytical approach. For example, the assumption that FAA is a phenomenon with a source in the frontal cortex is impossible to address using only a few frontal channel pairs. Using source localization (like DICS Beamforming used here) or source separation (for example, spatial filtering with generalized eigendecomposition/common spatial pattern [*Koles et al., 1990*; *Parra and Sajda, 2003*; *Tomé, 2006*]; or SPACE decomposition [*van der Meij et al., 2015*]; Roemer [*van der Meij et al., 2016*]) should be preferred when looking for the answer to this question. This point is important because frontal alpha sources are rarely measured reliably with EEG at the channel level: strong occipital and parietal alpha sources dominate alpha power recorded at frontal channels. As a result measuring FAA at the channel level could lead to poor signal to noise ratio and consequently to small effect size and low probability to observe a true FAA–DD relationship. On the other hand, if FAA does not originate in the frontal regions, it should be measured and interpreted differently. Such scenario is not unlikely because frontal alpha sources are generally difficult to detect with EEG/MEG. For example, Roemer *van der Meij et al., 2016*, using an advanced source separation method, found that 86.6% of the alpha components detected across subjects were occipito-parietal and only 1% (4/380) were frontal. If frontal alpha sources are difficult to detect using a source separation method designed to capture oscillatory sources, then it is likely that these sources are rarely observed at the channel level at all.

If FAA is not of frontal origin, then where could it come from? This is an open empirical question and we can only offer speculation here. *Jiang et al., 2016* have shown that the power of posterior alpha oscillations is reduced in depressed individuals and that this reduction strongly correlates with depression severity. Assuming that frontal projections from occipital or parietal sources will not be

**Table 10.** Results for analyses on data aggregated across studies, cluster-based permutation tests on frontal asymmetry in the source space.

Each row represents the result for given contrast × control for confounds combination. *N*: number of participants included in given contrast, *control for confounds*: whether the FAA data was residualized with respect to confounding variables (age, gender and education), *min t, max t*: lowest and highest t value in the search space, respectively; *n significant points*: total number of significant points in the search space before cluster-based correction; *n clusters*: number of clusters found in given analysis; *largest cluster p*: p-value for the largest cluster, *NA* means that no cluster was found in given analysis.

| No. | Contrast | N | Control for confounds | Aggregated source level analyses | | | | | |
|-----|----------|---|-----------------------|--------|--------|---------------------|------------|----------------------|---------------------|
| | | | | Min t | Max t | n significant points | n clusters | Largest cluster size | Largest cluster p |
| 1 | DvsHC | 102 vs 144 | − | −2.719 | 0.956 | 178 | 5 | 100 | 0.077 |
| 2 | DvsHC | 102 vs 143 | + | −2.846 | 0.908 | 211 | 5 | 116 | 0.066 |
| 3 | SvsHC | 77 vs 121 | − | −1.831 | −0.11 | 0 | 0 | NA | NA |
| 4 | SvsHC | 77 vs 120 | + | −1.871 | 0.182 | 0 | 0 | NA | NA |
| 5 | DReg | 102 | − | −2.387 | 1.376 | 31 | 3 | 25 | 0.328 |
| 6 | DReg | 102 | + | −3.095 | 1.354 | 112 | 3 | 68 | 0.155 |
| 7 | allReg | 315 | − | −3.044 | 0.411 | 174 | 6 | 160 | 0.067 |
| 8 | allReg | 314 | + | −3.107 | 0.282 | 235 | 6 | 195 | 0.054 |

DvsHC – diagnosed and healthy controls; SvsHC – sub-clinical and healthy controls; allReg – linear regression for all subjects together; DReg – linear regression for only diagnosed subjects.

perfectly symmetrical, a difference metric like FAA may be sensitive to posterior alpha power. Another possibility is that FAA originates from asymmetry at the source level; *Smith et al., 2018* demonstrated that channel-level FAA is related to source level asymmetry in frontal motor regions. However, the authors did not look for correlations with FAA in the full source space, but restricted their analyses to the R–L source level asymmetry and in consequence their analysis is insensitive to symmetrical sources of FAA. Our data and analyses cannot be conclusive in this regard because without a strong and reliable channel-level FAA–DD effect it is difficult to look for its source level correlate.

Tackling all the mentioned issues would require to systemize and unify the FAA methodology for the benefit of future studies. So far, *Kaiser et al., 2018a* proposed guidelines for methodology regarding subjects selection and controlling for confounding variables. *Smith et al., 2017* also suggested possible improvements in experimental procedures and EEG signal preprocessing. We believe there is still room for improvement in the signal analysis standards of FAA studies. Below we summarize our arguments and propose additional guidelines for EEG data analysis in FAA research:

- Always show the topography of the effects. Lack of topographical plots hinders interpretation in terms of both potential neural origin of the effect and its physiological reliability. It is a good idea to also add topographical plots of group averages: both for alpha power and alpha asymmetry (see *Figure 4—figure supplement 4*). Such visualizations can clarify the studied effect: when FAA is calculated as a R–L difference and is compared between groups, reasoning about difference between differences ((R–L) − (R–L)) can be unnecessarily complex.
- Conduct analysis on all frontal electrodes (or even all available electrodes) with correction for multiple comparisons. We recommend using the cluster-based permutation test (*Maris and Oostenveld, 2007*), as it is versatile and implemented in multiple software packages: mne-python (*Gramfort et al., 2013*; *Gramfort et al., 2014*) and fieldtrip (*Oostenveld et al., 2011*) for example, but the fieldtrip implementation is available also through EEGLAB (*Delorme and Makeig, 2004*) and brainstorm (*Tadel et al., 2011*).
- Try not to restrict the analysis to left minus right subtraction on averaged frequencies. As we show in the analyses on standardized data, avoiding subtraction and frequency averaging can uncover interesting effects that could otherwise be missed. Extending the search space to frequencies is straightforward when using the cluster-based permutation test.
- Perform analysis in the source-space if possible. Source localization allows to estimate the source of the signal more reliably and obtain a better signal to noise ratio (see, for example, *van Es and Schoffelen, 2019*). However, to minimize source localization error individual MRI

scans are required. Other methods focusing on source-separation like ICA, GED, or SPACE allow one to disentangle signal contributions from independent sources and increase signal to noise ratio. CSD was also proposed in this context in FAA literature before, but although it can mitigate some of the issues arising from volume conduction, it does not provide source localization or separation.
- Do not restrict the analysis to group contrasts if linear predictors are available. Using linear regression allows to take covariates into account and test hypotheses in a more detailed manner.

## Materials and methods

We included five data sets in the analyses. These data sets were obtained in five independent studies: Studies I–III have been collected by the authors of the article; Studies IV and V are publicly available: the data from Study IV were obtained from the PREDiCT repository (*Cavanagh et al., 2017*), while Study V data come from the MODMA database (*Cai et al., 2020*).

### Participants

In total 408 medication-free participants took part in the collected five studies: all without neurological disorders or head injuries. Thirteen subjects were excluded from further analyses due to excessive artifacts in the EEG signal (*Study II*: 8; *Study III*: 4; *Study IV*: 1) or missing data (*Study III:* 1; *Study IV:* 3). Additional three subjects were excluded from Study I because of not fulfilling the analysis criteria. As a consequence a total of 388 participants were included in the reported analyses: *Study I*, N = 51; *Study II*, N = 76; *Study III*, N = 91; *Study IV*: 117; *Study V*: 53. For descriptive statistics summarizing each study, see *Table 12* and *Figure 1*.

Participants in Studies II and III as well as healthy controls in Study I were recruited from the general population via advertisements in the local media or internal announcements for students at the University of Social Sciences and Humanities in Warsaw. In Study I diagnosed patients were recruited at the Psychiatry Clinic of the Department of Psychiatry, Medical University of Warsaw.

In Studies I–III each participant completed the Beck Depression Inventory (BDI) to determine the current level of mood disorder: we used BDI version I (*Beck et al., 1961*) in Studies I and II; and BDI version II (*Beck et al., 1996*) in Study III. Patients in Studies I and III were diagnosed with mild DD (F32.0) according to ICD-10 classification criteria after a structured clinical interview using the MINI – mini-international neuropsychiatric interview (*Sheehan et al., 1998*).

Participants in Study IV were recruited from the student population at University of Arizona. Participants with BDI score $\geq$13 were invited to participate in a Structured Clinical Interview for Depression. Participants meeting diagnostic criteria of current or past major DD were included in the group of diagnosed participants. Also, all participants completed BDI-II. More recruitment details can be found in the original papers (*Cavanagh et al., 2011*; *Cavanagh et al., 2019*).

In Study V participants with diagnosis of major DD were recruited from the Lanzhou University Second Hospital and healthy controls from the general population. In both groups, each participant completed the Patient Health Questionnaire (PHQ-9) (*Kroenke et al., 2001*) to evaluate depression level. More recruitment details can be found in the original papers (*Li et al., 2017*; *Sun et al., 2019*).

**Table 11.** Number of significant results compatible with the traditional FAA effect partitioned into analyses and contrasts.
If one or more such results have been found for a given cell, then the p-value for the binomial test is also shown.

| | Number of significant results congruent with the FAA effect | | |
| --- | --- | --- | --- |
| | Channel pairs N = 120 | Cluster correction N = 60 | Source space N = 30 |
| DvsHC | 2/32, p=0.48 | 1/16, p=0.56 | 2/8, p=0.057 |
| SvsHC | 0/24 | 0/12 | 0/6 |
| DReg | 0/32 | 0/16 | 0/8 |
| allReg | 0/32 | 0/16 | 0/8 |

**Table 12.** Descriptive statistics for each study presented in the article (N – number of participants, M – mean score, SD – standard deviation, BDI-I and BDI-II – Beck Depression Inventory I and II, PHQ-9 – Patient Health Questionnaire-9).

**Study I (N=51)**

|  | Diagnosed | Healthy Controls, BDI-I ≤ 5 | Subclinical | Unclassified |
|---|---|---|---|---|
| N | 29 | 22 | - | - |
| Age | M = 27.66, SD = 7.13 | M = 23.68, SD = 2.83 | - | - |
|  | 19 - 47 range | 20 - 33 range | - | - |
| Gender | 9 male, 20 female | 11 male, 11 female | - | - |
| BDI-I score | M = 20.93, SD = 8.21 | M = 2.00, SD = 1.48 | - | - |

**Study II (N=76)**

**Undiagnosed (N=76)**

|  | Diagnosed | Healthy Controls, BDI-I ≤ 5 | Subclinical, BDI-I ≥ 10 | Unclassified, 5 < BDI-I < 10 |
|---|---|---|---|---|
| N | - | 28 | 25 | 23 |
| Age | - | M = 25.32, SD = 6.46 | M = 24.44, SD = 5.08 | M = 25.22, SD = 6.78 |
|  | - | 18 - 43 range | 19 - 38 range | 18 - 40 range |
| Gender | - | 8 males, 20 females | 4 males, 21 females | 9 males, 14 females |
| BDI-I score | - | M = 2.29, SD = 1.72 | M = 17.56, SD = 8.13 | M = 7.91, SD = 1.16 |

**Study III (N=91)**

**Undiagnosed (N=64)**

|  | Diagnosed | Healthy Controls, BDI-II ≤ 5 | Subclinical, BDI-II ≥ 10 | Unclassified, 5 < BDI-II < 10 |
|---|---|---|---|---|
| N | 27 | 21 | 34 | 9 |
| Age | M = 27.19, SD = 7.23 | M = 24.29, SD = 4.99 | M = 25.06, SD = 6.58 | M = 26.78, SD = 8.74 |
|  | 19 - 42 range | 19 - 41 range | 18 - 44 range | 22 - 49 range |
| Gender | 6 males, 21 females | 7 males, 14 females | 10 males, 24 females | 2 males, 7 females |
| BDI-II score | M = 34.26, SD = 9.18 | M = 2.24, SD = 1.70 | M = 24.06, SD = 10.08 | M = 6.78, SD = 0.97 |

**Study IV (N=117)**

**Undiagnosed (N=95)**

|  | Diagnosed | Healthy Controls, BDI-II ≤ 5 | Subclinical, BDI-II ≥ 10 | Unclassified, 5 < BDI-II < 10 |
|---|---|---|---|---|
| N | 22 (12 past MDD, 10 present MDD) | 72 | 21 | 2 |
| Age | M = 18.91, SD = 1.34 | M = 19.00, SD = 1.23 | M = 18.43, SD = 0.81 | M = 18.00 |
|  | 18 - 24 range | 18 - 23 range | 18 - 21 range | ages 18, 18 |
| Gender | 8 males, 14 females | 33 males, 39 females | 3 males, 18 females | 1 males, 1 females |

*Table 12 continued on next page*

| BDI-II score | M = 21.82, SD = 5.70 | M = 1.60, SD = 1.48 | M = 22.95, SD = 4.25 | M = 6.50, SD = 0.71 |

| | Study V (N=53) | | | |
| --- | --- | --- | --- | --- |
| | Diagnosed | Healthy Controls, PHQ-9 ≤ 5 | Subclinical | Unclassified |
| N | 24 | 29 | - | - |
| Age | M = 30.88, SD = 10.37 | M = 31.45, SD = 9.15 | - | - |
| | 16 - 52 range | 19 - 52 range | - | - |
| Gender | 13 males, 11 females | 11 males, 13 females | - | - |
| PHQ-9 score | M = 18.33, SD = 3.50 | M = 2.66, SD = 1.80 | - | - |

In Studies I, III, and IV all undiagnosed participants (including the subclinical group) and diagnosed participants in Study I reported no past depression episodes. This information was not available for participants of Studies II and V.

Local ethics committees approved studies' protocols (Study I – the Medical University of Warsaw; Studies II and III – the University of Social Sciences and Humanities; Study IV – the University of Arizona; Study V – the Lanzhou University Second Hospital) and all participants signed consent forms.

## Electrophysiological data sets

The summary of all studies is presented in *Figure 1*. The equipment specifications and sessions details are provided below:

*Study I* – EEG signal was recorded with 64 channels (Ag/AgCl electrodes) arranged in the 10–5 system in a WaveGuard EEG Cap (Advanced Neuro Technology, ANT) at a sampling rate of 512 Hz. Impedance was kept below 10 kΩ. EEG signal was recorded during a 5-min session with eyes closed.

*Study II* – EEG signal was recorded with 64-Channel EGI HydroCel Geodesic Sensor Net, NetStation software, and an EGI Electrical Geodesic EEG System 300 amplifier at a sampling rate of 200 Hz. Impedance was kept below 40 kΩ. EEG signal was recorded during a 5-min session with eyes closed.

*Study III* – EEG signal was recorded with 64-Channel (Ag/AgCl active electrodes) Brain Products ActiCap system and BrainVision software at a sampling rate of 1000 Hz and downsampled off-line to 250 Hz. Impedance was kept below 10 kΩ. EEG signal was recorded during an 8-min session with alternating eyes open (O) and eyes closed (C) 1-min segments. The ordering of the segments was either OCCOCO or COOCOC (chosen randomly for each participant).

*Study IV* – EEG signal was recorded with 64-Channel (Ag/AgCl electrodes) Neuroscan Synamps[2] system at a sampling rate of 512 Hz. Impedance was kept below 10 kΩ. EEG signal contained six 1-min segments with alternating eyes open (O) and eyes close (C). The ordering of the segments was either OCCOCO or COOCOC.

*Study V* – EEG signal was recorded with 128-Channel EGI HydroCel Geodesic Sensor Net and NetStation software at a sampling rate of 250 Hz. Impedance was kept below 70 kΩ. EEG signal was recorded during a 5-min session with eyes closed.

## Data preprocessing

The preprocessing was performed with a custom-made EEGLAB-based MATLAB toolbox (eegDb: *Magnuski, 2020b*) and custom MATLAB scripts. Preprocessing steps were the same for all five studies. Continuous EEG signal was 1 Hz high pass filtered and divided into 1-s consecutive segments.

EEG recordings were visually inspected and segments containing strong or non-stereotypic artifacts were marked for rejection. These segments were ignored in all further preprocessing and analysis steps. Independent component analysis (ICA) was applied to remove remaining stereotypical artifacts from the data. Independent components signal, topographies, and power spectra were visually inspected and components related to eye blinks, eye movements, and muscular and cardiac artifacts (*Hipp and Siegel, 2013*; *McMenamin et al., 2010*; *Shackman et al., 2009*) were marked for removal. For extra safety the validity of component removal was also ensured by visually comparing the signal before and after ICA cleaning. The average number of removed components in each study were as follows: M = 7.20, SD = 3.80 in *Study I*; M = 8.29, SD = 3.50 in *Study II*; M = 9.41, SD = 5.49 in *Study III*, M = 3.36, SD = 2.62 in *Study IV*, M = 4.70, SD = 2.21 in *Study V*. Bad channels (*Study I*: M = 0.11, SD = 0.32; *Study II:* M = 0.72, SD = 1.04; *Study III:* M = 1.14, SD = 1.23; *Study IV*: M = 1.23, SD = 1.14; *Study V*: M = 0.81, SD = 0.92) were not included in the ICA and were interpolated after cleaning the signal with ICA. The signal was then re-referenced to common average (AVG) or CSD, depending on the type of analysis (see tables in *Results* section).

## Signal analysis

### Channel-pair and cluster-based analyses

All channel and source level analyses were performed using mne-python (*Gramfort et al., 2014*) and custom code (*Magnuski, 2020a*; *Magnuski, 2020c*; *Magnuski and Ruban, 2020*; all available on github). Half of the channel level analyses used CSD reference and the other half used average reference (AVG; see *Tables 3–6* for a summary). For each data set the continuous signal from eyes-closed rest period was used, starting 2 s after rest onset to avoid potential artifacts related to eyes closing. Power spectra were calculated using Welch method with 2 s long windows and a window step of 0.5 s. Welch windows overlapping with bad signal segments were removed and all remaining windows were averaged. This operation was performed for every channel and every subject giving rise to subjects by channels by frequencies matrix. Alpha asymmetry was calculated by first averaging spectral power in 8–13 Hz band, log-transforming and then for each left–right channel pair subtracting values obtained for left sites from those for right sites. We calculated alpha asymmetry as log(right)–log(left) because this is the most common approach.

### Cluster-based analyses on standardized data

When the right-side alpha pattern is topographically different from the left-side alpha pattern we cannot expect left vs right subtraction to reliably uncover alpha asymmetry. To alleviate this problem we performed an additional analysis that does not rely on subtraction. In this approach all frontal channels were used including those at the midline. Moreover, the alpha frequency range (8–13 Hz) was not averaged, and all frequency bins in this range were analyzed. Instead of right–left subtraction we standardized (z-scored) power in the selected channels by frequency space for each subject. Standardization should highlight asymmetry patterns that escape the traditional left vs right comparison, while also being sensitive to effects that do not rely on asymmetry at all.

### Source level analyses

Because channel-level projections can be highly variable depending on the source orientation we additionally perform analyses in the source space. We first digitized channel positions for a representative subject from Study III using photogrammetry. This step was performed because the default channel positions for many EEG caps assume a spherical head, which is not a realistic assumption for source localization. A hand-held video camera was used to record EEG cap placement on the head of the representative subject from multiple angles. The recorded video was processed with 3DF Zephyr (3DFlow 3DF Zephyr, Aerial Education version: *Toldo et al., 2015*) in order to obtain a 3D model of the subject's head and EEG cap. Channels positions' coordinates were extracted by manually placing control points on each channel in the 3d reconstruction. After coregistering the digitized channel positions with the fsaverage FreeSurfer head model (*Dale et al., 1999*, see next paragraph) we confirmed that the chosen subject's head shape was very similar to the fsaverage head model. These digitized channel positions (and thus the coregistration with the fsaverage) were used for all subjects in Studies I, III, and IV. As a result in Studies I and IV data from a few channels were not included in the source localization because these channels were not present in the created

digitization template. For Studies II and V default EGI channel positions were coregistered with the fsaverage model, because they assume a realistic head shape.

We employed Boundary Element Method (BEM) for the forward problem. We first created a three-layer (inner skull, outer skull, and outer skin) BEM model based on the FreeSurfer fsaverage template (*Dale et al., 1999*; *Fischl et al., 1999*). Next, the leadfield was constructed for a grid of 8196 equidistant source points covering the whole fsaverage cortical surface. Finally we used beamforming (Dynamic Imaging of Coherent Sources, DICS: *Gross et al., 2001*) to infer the source-level activity in alpha band.

The cross-spectral density matrices, necessary for DICS beamforming, were computed using Morlet wavelets (of length equal to seven cycles) on the continuous signal from the eyes-closed rest period starting from 2 s after rest onset. Bad signal segments were ignored, just like in the channel level analyses. To make the inverse solution more stable and noise-resistant we used a regularization parameter of 0.05 (*van Vliet et al., 2018*). Localized power maps were morphed to a symmetrical version of fsaverage brain (fsaverage_sym; *Greve et al., 2013*; *Van Veen et al., 1997*) to allow for left vs right comparisons. The asymmetry was computed in the same way as in the channel-pair and cluster-based analyses.

## Statistical analysis

We performed a multiverse analysis consisting of 270 analyses differing in: (a) the signal space used: channel space (average reference: 120 analyses, 44%, CSD reference: 120, 44%) or source space (DICS beamforming, 30, 11%); (b) subselection of the signal space: channel pairs (120, 44%), all frontal pairs with cluster correction (60, 22%) or all frontal channels with cluster-based correction and standardization instead of subtraction (60; 22%); (c) statistical contrast used: group comparisons or testing for a linear relationship (more information in the paragraph below); and (d) statistical control for confounding variables (135 without and 135 with control for confounds).

## Variants of statistical analysis

We used four different statistical contrasts in the analyses: two group contrasts and two linear contrasts. Group contrasts included: comparison between diagnosed and healthy controls (*DvsHC*) or sub-clinical and healthy controls (*SvsHC*). For group contrasts we used Welch t test, which does not assume equal variance of the compared groups (*Delacre et al., 2017*). Linear contrasts were performed either for all subjects together (*allReg*) or only for the diagnosed subjects (*DReg*).

These statistical contrasts are only used in the studies where they apply: for example, contrasting healthy and diagnosed subjects (*DvsHC*) cannot be done for Study II, where only healthy and sub-clinical participants are available. In the same way, comparing sub-clinical and healthy controls

**Table 13.** Summary of the contrasts (*DvsHC* – diagnosed vs healthy controls; *SvsHC* – sub-clinical vs healthy controls; *allReg* – linear regression between FAA and depression questionnaire score for all subjects together; *DReg* – linear regression only for the diagnosed subjects) and confounds (age, gender, and education) used in each study.

| | STUDY | | | | |
| | I | II | III | IV | V |
|---|---|---|---|---|---|
| **Contrast type** | | | | | |
| DvsHC | + | | + | + | + |
| SvsHC | | + | + | + | |
| allReg | + | | + | + | + |
| DReg | + | | + | + | + |
| **Control for confounds** | | | | | |
| gender | + | + | + | + | + |
| age | + | + | + | + | + |
| education | | + | + | | + |

(*SvsHC*) is not possible in Studies I and V, where only healthy and diagnosed participants are available. The availability of statistical contrasts in individual studies is summarized in *Table 13*. Whenever a contrast is available for a given study it is performed for all analysis spaces: average reference (AVG), CSD, and source level data.

For each statistical contrast and study we perform two data analysis approaches: (a) classical comparison of selected channel pairs and (b) cluster-based permutation test on the whole frontal asymmetry space. The channel pair analyses use two channel pairs, F3–F4 and F7–F8, or the corresponding channel pairs in the EGI cap (Studies II and V). The source space and standardized data analyses employ only the cluster-based approach.

Finally, all the analysis variants are performed twice: once in their standard form and the second time statistically controlling for potential confounding variables: gender, age, and education. These variables are added to the regression model explaining FAA, where the predictor of interest is either the depression status (*DvsHC* and *SvsHC* contrasts) or depression score in BDI/PHQ-9 questionnaire (*allReg* and *DReg* contrasts). The availability of these confounding variables in individual studies is shown in *Table 13*.

## Cluster-based permutation test

All the analyses that involve more than two selected channel pairs use cluster-based permutation tests (*Maris and Oostenveld, 2007*) to correct for multiple comparisons. Cluster-based permutation test is a nonparametric multiple comparison correction where the hypothesis of difference between conditions is evaluated at the level of multidimensional clusters. Clusters are formed by performing a chosen statistical test in the n-dimensional search space (channels or channels by frequencies in most of the analyses reported here) and grouping adjacent points where the test statistic passed some predefined threshold (typically an alpha level of 0.05). Each obtained cluster is then summarized by summing the statistics of all its members – that is, all adjacent points forming the cluster. These cluster summaries (cluster statistics) are then compared to a permutation null distribution of the maximum cluster statistic to obtain a p-value. The null distribution is approximated by a Monte-Carlo method where in each draw the condition labels are permuted between subjects (in this study: diagnosis status or BDI scores) and the statistical tests and clusters are computed in the same manner as for non-permuted data. As a result each Monte-Carlo draw produces cluster statistics from which the highest positive and the lowest negative value is saved. These values, when aggregated from all Monte-Carlo draws, constitute the null distribution for positive and negative effects to which cluster statistics from the actual analysis are compared.

For cluster-based analyses on standardized data, because they are sensitive to effects that do not have to be asymmetrical, significant test results were followed up with tests for asymmetry of the effects. For each cluster with p-value below 0.05 a chi-square test for two proportions was conducted comparing the proportion of cluster points on the left and right side of the head. Significant outcome of the test suggests that the cluster is asymmetrical.

Throughout all the analyses, including cluster-based permutation tests, we use an alpha level of 0.05. The same alpha level is used for cluster entry threshold in cluster-based tests. Results for single channel pairs, reported in *Table 1*, include also effect size (Cohen's d for group comparisons, Pearson's r for regression) and its 95% confidence interval calculated using bias-corrected accelerated bootstrap (*Tibshirani and Efron, 1993*; *Ho et al., 2019*).

## Analyses on aggregated data

To increase statistical power we perform additional analyses where we aggregate data across all studies that include identical contrasts. Because individual studies have different channel layouts these aggregated analyses are only performed when the studies can be mapped into a common space: (a) the selected channel pairs or (b) the source space. Before aggregation we tested the FAA values for selected channel pairs for equal variance across studies. Because the scale of the data can vary depending on lab-specific equipment or adopted impedance threshold, in case of unequal variance the FAA values were z-scored (centered and scaled) across participants within each study. The z-scoring was performed for channel pairs and source space analyses.

For aggregated channel pair data, we calculate the effect size (Cohen's d for group comparisons and Pearson's r for linear relationships) for each combination of contrast $\times$ channel pair $\times$ channel

space (AVG vs CSD) and estimate confidence intervals for the effect size using bias-corrected accelerated bootstrapping (*Tibshirani and Efron, 1993*; *Ho et al., 2019*). To estimate support for the claim of 'no effect' we calculate Bayes factor for the null hypothesis (BF01; *Rouder et al., 2009*) using the Pingoin python package (*Vallat, 2018*).

For aggregated source space data, we use cluster-based permutation tests. We do not estimate effect size and its confidence interval in source space because this would require a priori specification of a relatively narrow region of interest, which is not known. For all analyses using *allReg* or *DReg* contrasts the depression questionnaire scores are z-scored within each study. This is done because the aggregated studies use different questionnaires: BDI-I (Studies I and II), BDI-II (Studies III and IV), or PHQ-9 (Study V).

The aggregated analyses, like all other analyses, are performed twice: with and without statistical control for confounding variables. However, because not all confounding variables are available across studies, for each study we first explain the FAA data with the confounding variables using a regression model and then standardize and aggregate the model residuals.

## Acknowledgements

The work was supported by the grants from the Polish Ministry of Science and Higher Education: 'Diamond Grant' (DI2013012943), Iuventus Plus grant (0045/IP3/2011/71), N10601731/1344 grant, and from National Science Center: Preludium grant (2013/09/N/HS6/02890). Publication fee for this article was financed by the Ministry of Science and Higher Education in Poland under the 2019-2022 program „Regional Initiative of Excellence", project number 012/RID/2018/19. We thank Paweł Wroński for data collection in Study II; Paweł Holas and Dorota Wołyńczyk – Gmaj for psychiatric support in Studies I and III. Also, we thank James F Cavanagh and John JB Allen (PREDiCT data set) and Bin Hu with colleagues from Lanzhou University in China (MODMA data set) for making their data public which allowed us to use them in the analyses presented in this article. We acknowledge the support of COVID-19 pandemic in keeping us locked in homes with nothing more interesting to do but writing this manuscript.

## Additional information

### Funding

| Funder | Grant reference number | Author |
| --- | --- | --- |
| Ministerstwo Nauki i Szkolnictwa Wyższego | DI2013012943 | Aleksandra Kołodziej |
| Ministerstwo Nauki i Szkolnictwa Wyższego | 0045/IP3/2011/71 | Aneta Brzezicka |
| Ministerstwo Nauki i Szkolnictwa Wyższego | N10601731/1344 | Aneta Brzezicka |
| Narodowe Centrum Nauki | 2013/09/N/HS6/02890 | Mikołaj Magnuski |

The funders had no role in study design, data collection and interpretation, or the decision to submit the work for publication.

### Author contributions

Aleksandra Kołodziej, Conceptualization, Resources, Data curation, Formal analysis, Funding acquisition, Investigation, Visualization, Methodology, Writing - original draft, Project administration, Writing - review and editing; Mikołaj Magnuski, Conceptualization, Data curation, Software, Formal analysis, Visualization, Methodology, Writing - original draft, Writing - review and editing; Anastasia Ruban, Software, Formal analysis, Investigation, Visualization, Writing - original draft; Aneta Brzezicka, Conceptualization, Resources, Supervision, Funding acquisition, Methodology, Writing - review and editing

## Author ORCIDs
Aleksandra Kołodziej  https://orcid.org/0000-0002-6042-8215
Mikołaj Magnuski  http://orcid.org/0000-0001-6859-2581
Anastasia Ruban  https://orcid.org/0000-0001-7039-148X
Aneta Brzezicka  https://orcid.org/0000-0003-1950-4180

## Ethics
Human subjects: Ethical Review Boards approved studies' protocols: Study I - the Medical University of Warsaw; Studies II and III - the University of Social Sciences and Humanities (4/2017 and 25/2015 respectively); Study IV - the University of Arizona; Study V - the Lanzhou University Second Hospital. All participants signed informed consent forms.

## Decision letter and Author response
Decision letter https://doi.org/10.7554/eLife.60595.sa1
Author response https://doi.org/10.7554/eLife.60595.sa2

# Additional files

## Supplementary files
• Transparent reporting form

## Data availability
EEG data has been deposited to Dryad, and can be found under the https://doi.org/10.5061/dryad.5x69p8d18.

The following dataset was generated:

| Author(s) | Year | Dataset title | Dataset URL | Database and Identifier |
|---|---|---|---|---|
| Aleksandra K, Mikołaj M, Anastasia R, Aneta B | 2020 | No relationship between frontal alpha asymmetry and depressive disorders in a multiverse analysis of five studies | https://doi.org/10.5061/dryad.5x69p8d18 | Dryad Digital Repository, 10.5061/dryad.5x69p8d18 |

The following previously published datasets were used:

| Author(s) | Year | Dataset title | Dataset URL | Database and Identifier |
|---|---|---|---|---|
| Cavanagh JF, Allen JB | 2017 | Depression Rest | http://predict.cs.unm.edu/downloads.php | PREDiCT, d006 |
| Cai H, Gao Y, Sun S, Li N, Tian F, Xiao H, Li J, Yang Z, Li X, Zhao Q, Liu Z, Yao Z, Yang M, Peng H, Zhu J, Zhang X, Gao G, Zheng F, Li R, Guo Z, Ma R, Yang J, Zhang L, Hu X, Li Y, Hu B | 2015 | EEG_128channels_resting_lanzhou_2015 | http://modma.lzu.edu.cn/data/application/ | MODMA, EEG_128channels_resting_lanzhou_2015 |

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
