## [Decision Letter]

**Acceptance summary:**

The authors leveraged a multiverse approach to investigate the validity and robustness of frontal EEG alpha asymmetry as a depression biomarker. In total, 270 analyses were performed on data from 5 independent samples, providing one of the largest and most comprehensive studies in the literature.

**Decision letter after peer review:**

Thank you for submitting your article "No relationship between frontal alpha asymmetry and depressive disorders in a multiverse analysis of three studies" for consideration by *eLife*. Your article has been reviewed by 2 peer reviewers and the evaluation has been overseen by Alexander Shackman as the Reviewing Editor and Chris Baker as the Senior Editor.

Based on a discussion with the 2 Reviewers, the Reviewing Editor has drafted this decision to help you prepare a revision.

Summary:

Authors report results from 81 alternative analysis paths for resting frontal asymmetry data recorded in three independent studies (N = 51, N = 76, N = 91).

The Reviewers and I saw several strengths in the manuscript:

• Multiverse approach

• Methodological rigor

• Transparent and comprehensive methods reporting

• Sound EEG methods

• Publicly available code

• With appropriate revision, could be a useful contribution to the frontal alpha asymmetry (FAA) literature

Nonetheless, we had some important reservations about the manuscript in its present form.

Essential revisions:

1. Power.

a. The sample sizes in each of the three studies are much too small to derive any firm conclusions concerning the relationship between alpha asymmetry and depressive disorders. Whereas the authors acknowledge that their samples are too small to reliably detect small to moderate effect (p. 21), this does not really solve the problem, given that there is no reason to expect a large effect based on the more recent findings and meta-analyses that the authors cite. As it stands, each of the studies reported is simply underpowered for the effect of interest and this problem cannot be overcome by computing a larger number of alternative analyses.

b. This work could still be a valuable contribution if the focus were shifted towards a comparison of different defensible analysis paths for FAA, and documenting the surprisingly low degree of convergence between them, and providing useful open scripts and tools along with the authors' fitting guidelines for EEG data analysis in FAA research. If the authors considered this a reasonable way to move forward, interesting additions to the current version may include a correlation matrix of the asymmetry scores derived with different analysis paths, histograms of the t-values of the final statistical test of interest across analysis options (obviously this will not be possible for all analysis options), and a somewhat more complete documentation of the analysis scripts on GitHub. Alternatively, the authors could also apply their multiverse analysis to a different data set featuring sufficient statistical power to detect small-moderate effects (such data sets exist and it might even be possible to use them for that purpose).

2. Psychiatric Differences Between the Present Report and the Published Literature. (Noted by Multiple Reviewers). To the degree that the authors choose to focus the revision on the issue of FAA and Depression, they need to do a better job acknowledging potentially important differences between the present study and prior work.

a. The authors report their currently depressed sample was diagnosed as "Mild Depression Disorder". How does that diagnosis (MildDD) compare to earlier studies that used participants with MajorDD, in terms of being predicted by FAA? At least one earlier study found that lifetime Major Depression and not current depression was predicted by FAA using groups that were several times the sample size of the current study (Stewart et al., 2010). Thus, current BDI score might not be predictive if the depressed groups have few or brief depressive episodes. Given the above it might not be surprising that the only significant finding in the current study came from comparing a diagnosed depressive group to a healthy control group in one of the studies (Study I). Thus, expectations for lower differences among those in the sub-clinical range and controls in Studies II and III makes sense and suggests depression severity might be an important factor. At minimum, it would be useful for the authors to report more information about single episode vs. chronic/recurrent MDD.

b. The authors provide no assessment of lifetime psychiatric history. Lifetime MDD has sometimes been associated with FAA. If some of the mild or subclinical cases have a lifetime history of MDD, this would tend to lessen differences with the current MDD group. I would encourage the authors to, at minimum, raise this possibility in their report.

c. Gender. Many published FAA-MDD studies have relied on predominantly or entirely female samples. A few have reported differences when analyzing men in their groups (e.g., Miller et al., 2002). Stewart et al. (2010) found gender differences in the patterns of FAA predicting depression. Even van der Vinne et al. (2017) found a significant interaction with gender. There has been the suggestion that if men show an opposite pattern of FAA, and gender is not considered during analyses, men and women might cancel-out each other's effects. Thus, gender might be important to consider in this study since among the 8 groups assessed, men made-up from half to about a quarter of the groups. I am not suggesting that the authors conduct their 81 analyses a second time, taking gender into account (particularly given the underpowered nature of the samples). However, I think it would be useful for there to be some statement of whether gender differences in BDI scores and FAA are evident. And a statement in the Discussion about what possible gender influences might or might not mean for their results, and for the study of FAA-MDD.

3. BDI Analyses.

a. Restricting the linear regression analyses of frontal asymmetry on BDI to only the depressive groups in studies I and II (No. 11-14 in Tables 2, 3 and 4, No. 6-7 in Table 5) and to only the non-depressive group in study III (No. 17 and 18 in Tables 2, 3 and 4, N. 9 in Table 5) does not seem to be a defensible analysis approach, given that the largest difference is theoretically expected between depressed and non-depressed participants. We would encourage the authors to omit these analyses.

---

## [Author Response]

Essential revisions:1. Power.a. The sample sizes in each of the three studies are much too small to derive any firm conclusions concerning the relationship between alpha asymmetry and depressive disorders. Whereas the authors acknowledge that their samples are too small to reliably detect small to moderate effect (p. 21), this does not really solve the problem, given that there is no reason to expect a large effect based on the more recent findings and meta-analyses that the authors cite. As it stands, each of the studies reported is simply underpowered for the effect of interest and this problem cannot be overcome by computing a larger number of alternative analyses.

We agree that each of the studies alone is underpowered. We now included two additional datasets in our analyses: one from the PREDiCT Repository (predict.cs.unm.edu) and one from the Modma Dataset (modma.lzu.edu.cn). This (and other modifications explained later in this response) raised the number of analyses from 81 to 270 and the total number of participants from 218 to 388. This step on its own does not mitigate the problem of single studies being underpowered – the included datasets contain: for PREDiCT dataset (Study IV) – 117 subjects in total but only 22 depressed are available; and for in Modma dataset (Study V) – 53 subjects in total, 24 depressed. We therefore additionally conduct a set of analyses on data aggregated across studies (Figures 8 and 9). The channel level analyses on aggregated data were present in our original submission in supplementary materials – we now place these updated results in the main text. Due to the added two studies we are now able to better estimate the confidence intervals for the effect size and can exclude also moderate FAA effects. We also show that almost all Bayes factors for the null hypothesis (BF01) are in the 3 – 10 range in the aggregated analyses which means moderate support for the null hypothesis. All results with BF01 values below or close to 3 point in the direction incompatible with the classical FAA effect. We also aggregate the datasets in the source space and perform cluster-based tests: although these results are not nominally significant they are in the conventional “trend” range (< 0.1). We find it interesting that the source level analyses are more consistent in their direction with the classical FAA effect compared to the analyses in sensor space. We now discuss this in the text and express more hope for future studies employing source space analyses.

b. This work could still be a valuable contribution if the focus were shifted towards a comparison of different defensible analysis paths for FAA, and documenting the surprisingly low degree of convergence between them, and providing useful open scripts and tools along with the authors' fitting guidelines for EEG data analysis in FAA research. If the authors considered this a reasonable way to move forward, interesting additions to the current version may include a correlation matrix of the asymmetry scores derived with different analysis paths, histograms of the t-values of the final statistical test of interest across analysis options (obviously this will not be possible for all analysis options), and a somewhat more complete documentation of the analysis scripts on GitHub. Alternatively, the authors could also apply their multiverse analysis to a different data set featuring sufficient statistical power to detect small-moderate effects (such data sets exist and it might even be possible to use them for that purpose).

As we mentioned above, we decided to obtain additional datasets and emphasize more the analyses on aggregated data. We are also updating the code documentation on GitHub and adding short examples on how to replicate some of the analyses presented in the paper and how to include additional datasets in our analysis pipeline. We believe that these changes make our analyses even more transparent and reusable.

2. Psychiatric Differences Between the Present Report and the Published Literature. (Noted by Multiple Reviewers). To the degree that the authors choose to focus the revision on the issue of FAA and Depression, they need to do a better job acknowledging potentially important differences between the present study and prior work.a. The authors report their currently depressed sample was diagnosed as "Mild Depression Disorder". How does that diagnosis (MildDD) compare to earlier studies that used participants with MajorDD, in terms of being predicted by FAA? At least one earlier study found that lifetime Major Depression and not current depression was predicted by FAA using groups that were several times the sample size of the current study (Stewart et al., 2010). Thus, current BDI score might not be predictive if the depressed groups have few or brief depressive episodes. Given the above it might not be surprising that the only significant finding in the current study came from comparing a diagnosed depressive group to a healthy control group in one of the studies (Study I). Thus, expectations for lower differences among those in the sub-clinical range and controls in Studies II and III makes sense and suggests depression severity might be an important factor. At minimum, it would be useful for the authors to report more information about single episode vs. chronic/recurrent MDD.b. The authors provide no assessment of lifetime psychiatric history. Lifetime MDD has sometimes been associated with FAA. If some of the mild or subclinical cases have a lifetime history of MDD, this would tend to lessen differences with the current MDD group. I would encourage the authors to, at minimum, raise this possibility in their report.

The two additional datasets included in the current version of manuscript contain individuals diagnosed with major depressive disorder. But we agree that it is important to describe potential differences with respect to prior studies. We added this information – to the extent that it is available – to the description of participants in the manuscript.

In Study III some of the MildDD participants had a history of past depression, however all other participants (including subclinical group) reported no past depression episodes. In Study IV both past and present MajorDD participants are included but are not numerous enough to analyse them separately. Moreover, the past and present groups do not differ in BDI scores (t = -0.648, p = 0.524) so these groups are merged in our analyses. In Study IV participants belonging to the control group did not report past episodes of depression. In Study I both diagnosed and healthy participants did not report past episodes of depression. Unfortunately we do not have this information available for the other studies (II and V). However, we would like to emphasize that for the two datasets included earlier in the analyses (Study I and III) diagnosed participants expressed high BDI scores – some higher than the majorDD patients in Study IV for example (see the histograms in Figure 1). In general, we believe that if a depressed participant had enough motivation and self-control to make an appointment for the EEG recording and then show up in the lab around the appointed time then it is likely that this participant has currently lower depression severity or could be classified as MildDD by another clinicist.

Although we retain the subclinical vs controls contrast in our analyses (because some previous studies defined depressed and control groups solely on the basis of psychometric score) we tried to emphasise the depressed vs healthy contrast (DvsHC) more.

c. Gender. Many published FAA-MDD studies have relied on predominantly or entirely female samples. A few have reported differences when analyzing men in their groups (e.g., Miller et al., 2002). Stewart et al. (2010) found gender differences in the patterns of FAA predicting depression. Even van der Vinne et al. (2017) found a significant interaction with gender. There has been the suggestion that if men show an opposite pattern of FAA, and gender is not considered during analyses, men and women might cancel-out each other's effects. Thus, gender might be important to consider in this study since among the 8 groups assessed, men made-up from half to about a quarter of the groups. I am not suggesting that the authors conduct their 81 analyses a second time, taking gender into account (particularly given the underpowered nature of the samples). However, I think it would be useful for there to be some statement of whether gender differences in BDI scores and FAA are evident. And a statement in the Discussion about what possible gender influences might or might not mean for their results, and for the study of FAA-MDD.

We agree that control for confounding variables should be performed more often in FAA studies. After reflection, we have decided to conduct all our analyses twice: with additional analyses taking confounding variables (gender, age and education) into account. These results are included as tables, referenced in the text and some of the figures. This issue is also now referenced in the introduction.

We’ve confirmed that we see no significant BDI differences between diagnosed male and female participants: Study I: t = 0.332, p = 0.745

Study III: t = 0.413, p = 0.694

Study IV: t = -0.957, p = 0.356

Study V: t = 0.818, p = 0.426 (PHQ-9)

3. BDI Analyses. (Noted by Multiple Reviewers).a. Restricting the linear regression analyses of frontal asymmetry on BDI to only the depressive groups in studies I and II (No. 11-14 in Tables 2, 3 and 4, No. 6-7 in Table 5) and to only the non-depressive group in study III (No. 17 and 18 in Tables 2, 3 and 4, N. 9 in Table 5) does not seem to be a defensible analysis approach, given that the largest difference is theoretically expected between depressed and non-depressed participants. We would encourage the authors to omit these analyses.

As suggested by the reviewers, we’ve now removed the regression analyses on non- depressed participants (nonDreg contrast). We decided to retain the DReg contrast (regression on diagnosed subjects) for the following reasons:

– When looking for a linear relationship between two variables in the presence of group differences linear regression can recapitulate the group differences even though no linear relationship is present. Therefore to demonstrate linear relationship it is important to show it in at least one of the subgroups (see for example Makin and de Xivry, 2019).

– A marker that is sensitive to depression severity would be more clinically relevant as it could be used for example to track the progress during therapeutic or pharmacological interventions. As a consequence FAA would be a useful marker even in the absence of group differences in DvsHC contrast.

However we tried to emphasise the DvsHC contrast more in the text. We also provide a separate table in the discussion calculating binomial test separately for contrasts and analysis types.

When removing the nonDReg contrast and investigating BDI/PHQ-9 histograms for the studies we decided to extend the allReg test to Studies I, IV and V because in all these studies depressed participants show a wide range of psychometric scores and including more participants in a single regression might have more statistical power than t test on extreme groups.

Reference:

Makin, T. R., and de Xivry, J.-J. O. (2019). Ten common statistical mistakes to watch out for when writing or reviewing a manuscript. In eLife (Vol. 8). https://doi.org/10.7554/eLife.48175